# Actomyosin-mediated apical constriction promotes physiological germ cell death in *C. elegans*

Tea Kohlbrenner[1,2], Simon Berger[1,3], Ana Cristina Laranjeira[1,2], Tinri Aegerter-Wilmsen[1], Laura Filomena Comi[1,2], Andrew deMello[3], Alex Hajnal[1] *

1 Institute of Molecular Life Sciences, University of Zürich, Zürich, Switzerland, 2 Molecular Life Science PhD Program, University of Zürich and ETH Zürich, Zürich, Switzerland, 3 Institute for Chemical and Bioengineering, ETH Zürich, Zürich, Switzerland

* alex.hajnal@mls.uzh.ch

## Abstract

Germ cell apoptosis in *Caenorhabditis elegans* hermaphrodites is a physiological process eliminating around 60% of all cells in meiotic prophase to maintain tissue homeostasis. In contrast to programmed cell death in the *C. elegans* soma, the selection of germ cells undergoing apoptosis is stochastic. By live-tracking individual germ cells at the pachytene stage, we found that germ cells smaller than their neighbors are selectively eliminated through apoptosis before differentiating into oocytes. Thus, cell size is a strong predictor of physiological germ cell death. The RAS/MAPK and ECT/RHO/ROCK pathways together regulate germ cell size by controlling actomyosin constriction at the apical rachis bridges, which are cellular openings connecting the syncytial germ cells to a shared cytoplasmic core. Enhancing apical constriction reduces germ cell size and increases the rate of cell death while inhibiting the actomyosin network in the germ cells prevents their death. We propose that actomyosin contractility at the rachis bridges of the syncytial germ cells amplifies intrinsic disparities in cell size. Through this mechanism, the animals can adjust the balance between physiological germ cell death and oocyte differentiation.

## Introduction

Programmed cell death, commonly called apoptosis, is an evolutionary conserved process that is essential for the development, morphogenesis, and survival of most multicellular organisms [1–4]. During *Caenorhabditis elegans* embryonic and larval development, an invariant set of 131 somatic cells is eliminated through programmed cell death as part of normal development [5–10]. In addition, in adults grown under standard conditions, around 60% of the germ cells at the pachytene stage of meiotic prophase I are eliminated by apoptosis before they can complete oogenesis [11–13]. Germ cell corpses are rapidly engulfed and digested by the somatic sheath cells that form the walls of the tubular gonad arms [11,14,13]. Unlike the programmed death of somatic cells, the death of meiotic germ cells in adult hermaphrodites occurs randomly. The elimination of healthy germ cells by apoptosis is a physiological process that is

**Data Availability Statement:** All relevant data are within the paper and its Supporting Information files. Numerical data are included in S1 Data table,

reagents and plasmids used are listed in the S2 and S3 Data tables, scripts used for image processing can be found at the end of the supplementary information file, and uncropped Western blots used for quantification are shown in S1 Raw Images.

**Funding:** This project was supported by the Kanton Zürich, the ETH Zürich, Candoc grant no. FK-20-090 from the Forschungskredit of the University of Zürich to T.K. (www.research.uzh) and by the Swiss National Science Foundation grant no. 31003A- 166580 to A.H. (www.snf.ch). The funders had no role in study design, data collection and analysis, decision to publish, or preparation of the manuscript.

**Competing interests:** The authors have declared that no competing interests exist.

**Abbreviations:** DTC, distal tip cell; GEF, Guanine Nucleotide Exchange Factor; InsR, insulin receptor; MLC, myosin light chain; NGM, nematode growth medium; RNAi, RNAi interference.

thought to remove excess cells to maintain tissue homeostasis and redistribute resources among the surviving germ cells [11,15]. Programmed, somatic, and physiological germ cell death utilize the same core cell death (CED) machinery to execute apoptosis via CED-3 caspase activation, except for the EGL-1 apoptosis activator used only in the soma [5]. However, the signals by which individual germ cells are selected to die have so far remained unknown.

The gonads of *C. elegans* hermaphrodites are formed by 2 U-shaped tubes each connected to a common uterus [16]. One gonad arm in adult animals contains around 1,000 germ cells arranged in a distal to proximal polarity (**Fig 1A**) [17]. Germ cells in the distal gonad and the loop region form a large syncytium, in which the germ cell nuclei are only partially enclosed by plasma membranes. Each syncytial germ cell is connected on its apical side through an opening, called rachis bridge, to a common cytoplasmic core, the rachis (**Fig 1A' and 1A"**) [16]. The rachis bridges are lined with contractile actomyosin rings, which can constrict to close the openings to the rachis, thereby regulating the exchange of cytoplasm between germ cells and rachis [18,19]. The size of the rachis bridges is dynamically regulated according to the meiotic stage of the cells. The rachis bridges first constrict as cells pass through the mid to late pachytene region, after which they enlarge in cells that exit pachytene and enter diakinesis/diplotene in the loop region until they are fully constricted to cellularize the maturing oocytes [19].

Signals transduced by the RHO family of small GTPases regulate cell shape in a variety of processes, including gastrulation, cytokinesis, cell migration, and epithelial morphogenesis. In most of these processes, the activation of RHO signaling by extracellular or cell-intrinsic signals induces the constriction of the cortical actomyosin network to generate intracellular forces, which alter cell shape [20–22]. One of the central downstream effectors is the RHO-dependent kinase ROCK, which induces actomyosin constriction by phosphorylating the regulatory myosin light chain (MLC) subunit. In the *C. elegans* germline, actomyosin contractility is regulated by the RHO Guanine Exchange Factor (GEF) ECT-2 and the RHO-1 small GTPase that activates the ROCK homolog LET-502, which phosphorylates the myosin regulatory light chain MLC-4 [23–25]. Phosphorylated, activated MLC-4 in a complex with the essential light chain MLC-5 and the myosin heavy chain NMY-2 induces constriction of the F-actin cytoskeleton [26]. MLC-4 can also be activated through a RHO/ROCK-independent pathway composed of the p21-activated kinase PAK-1 and its activator PIX-1 [26–28].

Germline stem cells in the distal-most mitotic zone are induced by a DELTA/NOTCH signal from the distal tip cell (DTC) to proliferate (**Fig 1A**) [17]. As the germ cells migrate proximally, they enter the pachytene stage of meiotic prophase I. While progressing through the pachytene region, germ cells receive external signals via the DAF-2 insulin receptor (InsR), which activates the RAS/MAPK pathway [29]. RAS/MAPK signaling is necessary not only for the surviving germ cells to exit pachytene and differentiate into mature oocytes but also for germ cell death, which occurs almost exclusively in the mid to late pachytene region (**Fig 1A**) [11,13,30,31]. The mechanisms, by which MAPK activation triggers the apoptosis of individual germ cells, remain unknown.

Here, we show that RAS/MAPK signaling is necessary for NMY-2 myosin enrichment at the rachis bridges to promote apical germ cell constriction and reduce germ cell size. Smaller germ cells are then selectively eliminated by apoptosis and donate their cytoplasm to surviving germ cells, which grow in size [32,33]. Based on these findings, we propose that global actomyosin contractility in the syncytial germline, determined by the joint activities of the RAS/MAPK and ECT/RHO/ROCK signaling pathways, determines the rate of physiological germ cell death.

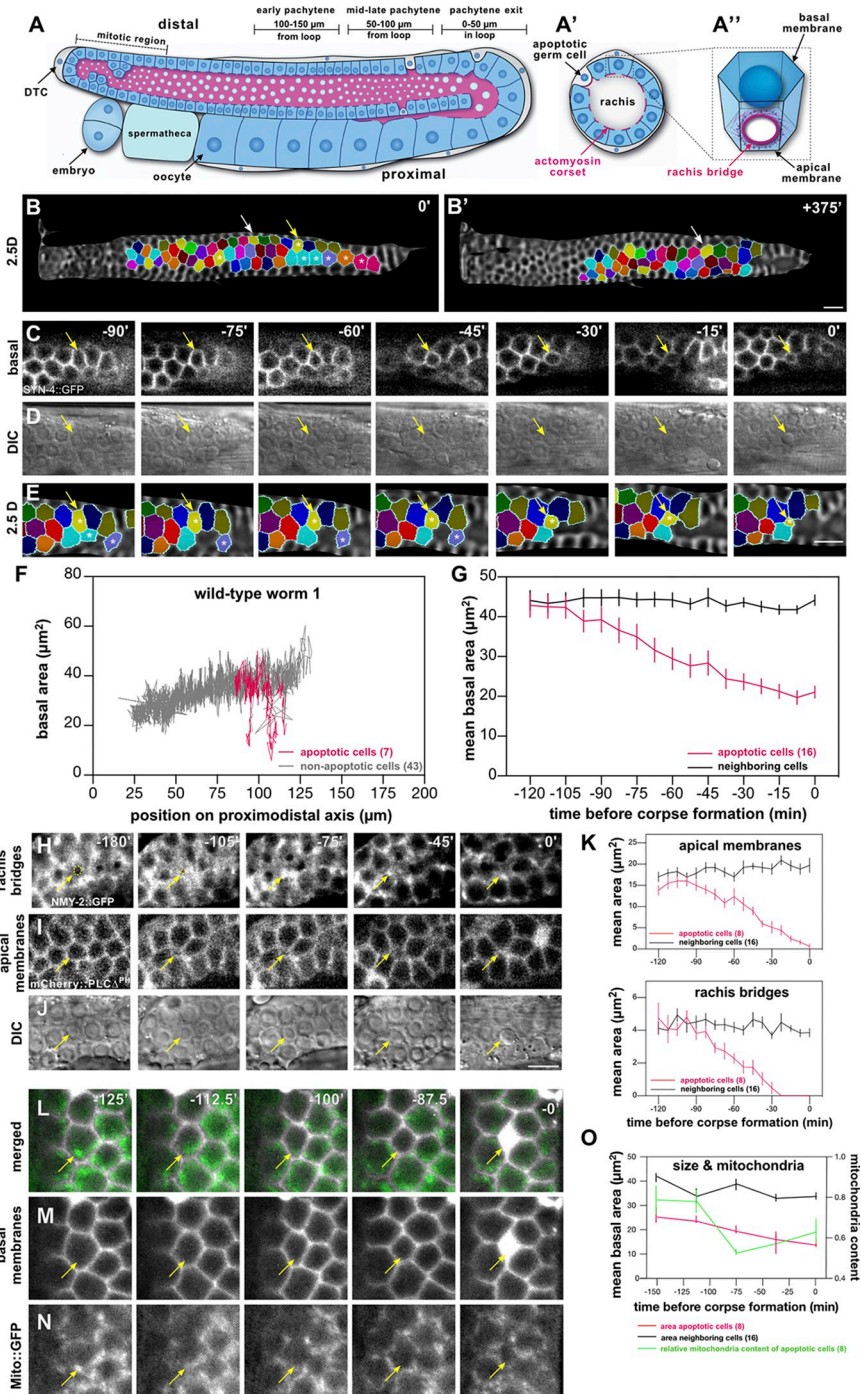

**Fig 1. Germ cell size decreases before corpse formation.** (**A**) Schematic drawing of a single gonad arm, oriented with its distal end on the top left and the proximal end on the bottom left side, with (**A'**) showing a cross-section through the pachytene region and (**A"**) a single germ cell with its rachis bridge. Live germ cells are indicated with a blue cytoplasm, while the cytoplasm of apoptotic germ cells and the somatic sheath cells are colored gray. The actomyosin corset lining the apical germ cell membranes and the rachis bridges is shown in magenta. The 3 zones used for measurements in the following figures are indicated as "0–50 μm in the loop" (pachytene exit of survivors), "50–100 μm from the loop" (mid to late pachytene), and "100–150 μm from the loop" (early pachytene). (**B**) Start (0 minutes) and (**B'**) endpoint (+375 minutes) of a germ cell tracking experiment in the wild type showing the color-coded germ cells in 2.5D projections of the basal membrane surface. See S1 Movie for all time frames. The yellow arrow in (**B**) points to a preapoptotic germ cell that was tracked as shown in the magnified images in (**C-E**), while the

white arrows in (**B**) and (**B'**) point to a surviving germ cell. The white asterisks highlight other cells that underwent apoptosis during the recording. (**C**) Single z-sections, (**D**) the corresponding DIC images, and (**E**) the segmented cells on the 2.5D projections. (**F**) Basal areas of cells were tracked in a single wild-type animal and plotted against their relative positions along the distal to proximal axis. The traces of apoptotic cells are shown in magenta, and those of surviving cells or of cells that did not die during the recording are shown in gray. Therefore, small germ cells in Figs 1F, S1A, and S1B that could not be tracked until corpse formation are also labeled in gray. See S1A and S1B Fig for traces of 2 additional animals. (**G**) Mean basal cell area ±SEM of apoptotic germ cells tracked in 3 wild-type animals (magenta line) and their neighboring cells (black line) plotted against the relative time before an apoptotic cell corpse was first detected in the DIC channel. Neighboring cells were defined as cells within 16 μm (corresponding to approximately 2 cell diameters) from the center of the apoptotic cells. (**H**) Time-lapse observation of the rachis bridges outlined with the NMY-2::GFP reporter, (**I**) the apical plasma membranes labeled with the mCherry::PLCΔ$^{PH}$ reporter, and (**J**) the corresponding DIC images in wild type. For each channel, a single z-section is shown. The yellow arrows point to a cell undergoing apoptosis. The yellow dashed circles in (**H**) outline the rachis bridge of the apoptotic cell. (**K**) Mean area ±SEM of the rachis bridges and the apical plasma membrane in apoptotic cells (magenta lines) and their left and right neighbors (black line), plotted against the relative time before corpse formation. A mean rachis bridge area of 0 indicates full constriction. (**L**) Time-lapse observation of the basal plasma membranes labeled with the mCherry::PLCΔ$^{PH}$ (gray) and the Mito::GFP (green) reporters. The yellow arrows point to a germ cell undergoing apoptosis. (**M, N**) The separate channels. (**O**) Mean area ±SEM of the basal area in apoptotic cells (magenta lines), their left and right neighbors (black line), and the relative mitochondria content ±SEM in apoptotic cells (green line), plotted against the relative time before corpse formation. Mitochondria content in apoptotic cells was quantified as the variation in the granular Mito::GFP signal intensity inside the cell perimeter and normalized to the variation in the neighboring nonapoptotic cells (see extended methods in S1 File). In each graph, the numbers in brackets refer to the total number of cells analyzed. See S1 Data for the underlying data and results of the statistical analysis. The scale bars are 10 μm.

## Results

### Germ cell size decreases before corpse formation

To observe germ cells undergoing apoptosis, we used a custom-made microfluidic device, which allowed us to perform long-term imaging and follow the fates of individual germ cells under physiological conditions [12]. Germ cells in the pachytene stage of meiotic prophase I were tracked over 7 hours in young adult wild-type animals starting 72 hours after the L1 stage, using the *Psyn-4>syn-4::gfp* membrane marker (*xnIs87*) to outline the cell borders and DIC optics to visualize apoptotic corpses. Image stacks were acquired every 7.5 minutes, and 2.5D projections of the basal germ cell surfaces on a curved mesh around the gonads were created using MorphographX software [34] (**S1C Fig** and the **extended methods** in **S1 File**). The basal projections of the consecutive time points were manually aligned, such that individual germ cells could be color-coded and tracked over time (**Fig 1B–1E** and **S1 Movie**).

Analysis of a total of 155 germ cells tracked in 3 animals revealed that the basal surface area of germ cells undergoing apoptosis decreased prior to the appearance of a cell corpse in the DIC image (time-point 0' in **Fig 1D**), whereas the surviving neighboring cells increased in size as they moved proximally (**Figs 1F**, S1A, **and** S1B). Relative to their surviving neighbors, cells fated to die started to shrink 1 to 2 hours before apoptotic corpses became visible (**Fig 1G**). The average basal area of the 16 apoptotic germ cells that could be tracked until corpse formation was $21 \pm 6$ μm$^2$ at the last time frame before they were eliminated. When the basal area dropped below 20 μm$^2$, cells formed a corpse within an hour with a probability of 58%, compared to a probability of 1% for cells with a basal area greater than 20 μm$^2$. Therefore, a basal area below 20 μm$^2$ may serve as a predictor for imminent apoptosis. By creating projections of increasing depths, we found that the basal and apical membrane domains of preapoptotic cells decreased simultaneously, while their height did not increase, indicating a reduction in the total volume of preapoptotic germ cells (**S1D and S1E Fig**).

Next, we used an endogenous reporter for the nonmuscle myosin *nmy-2* (*nmy-2::gfp(cp13)*) [35] to visualize the rachis bridges on the apical side of the syncytial germ cells together with a *pie-1>mCherry::PLCΔ$^{PH}$* membrane reporter (*itIs37*) to label the plasma membranes. The apical actomyosin rings were fully constricted 30 to 45 minutes before corpse formation, leading

to the full cellularization of apoptotic cells before corpse formation (**Fig 1H–1K**). Cellularized germ cells were characterized by a round rather than a honeycomb shape on their basal surface (see the -45' time-point in **Fig 1C**).

One of the earliest signs of germ cell death is the expulsion of mitochondria from the preapoptotic germ cells [32]. To determine if the reduction of germ cell size occurs at the onset of apoptosis or later during the execution phase, we correlated germ cell size with mitochondrial content. We therefore tracked preapoptotic germ cells in animals expressing a *pie-1>mito*::*gfp* mitochondria marker (*mjSi68*) together with the *pie-1>mCherry*::*PLCΔ^{PH}* membrane reporter (*itIs37*) and quantified the basal area as well as the Mito::GFP content in shrinking germ cells relative to their nonapoptotic neighbors (**Fig 1L–1N**). This analysis indicated that germ cell shrinkage and loss of mitochondria occur simultaneously (**Fig 1O**). Moreover, cell size quantification in static images of the pachytene region showed a strong correlation between small germ cell size and the absence of mitochondria (**S1E Fig**). Thus, the size reduction of dying germ cells occurs at the onset of apoptosis.

Taken together, the long-term tracking experiments indicated that germ cells fated to die undergo apical constriction, reduce their size, and lose their mitochondria 1 to 2 hours before forming apoptotic corpses that are engulfed by the somatic sheath cells. However, it should be noted that these observations could not distinguish if the reduction in germ cell size is a consequence or a cause of the apoptotic fate.

## CED-3 is necessary for the formation of small germ cells

To investigate the influence of the CED pathway (cell death abnormal) on germ cell size, we tracked germ cells in apoptosis-deficient *ced-3(n717lf) caspase* mutants [36] using the same approach as for the wild type (**Fig 1B**). Data for 176 germ cells tracked in 3 animals indicated that no germ cells that were consistently smaller than their neighbors were present in the pachytene region of *ced-3* loss-of-function (*lf*) mutants (**S2 Movie**). Cell size measurements (**S1C Fig**) confirmed the loss of the small cell population (basal cell area below 20 μm$^2$) in *ced-3(lf)* mutants, especially in the mid to late pachytene region (50 to 100 μm from the loop), where most apoptotic cell death occurs in the wild type (**Figs 2A, 2B, S2A, and S2B**).

We further examined the localization of actomyosin network components in apoptotic corpses using endogenous reporters for the RHO GEF ECT-2 and the RHO kinase LET-502, along with the nonmuscle myosin NMY-2 marker and a germline-specific LifeAct F-actin reporter. All these actomyosin pathway components were strongly enriched at the cortex of dying cells (**Fig 2C–2F'**). Possibly, the overall size reduction of dying cells may have contributed to the compaction of the actomyosin network.

Taken together, these data suggest that the full activation of the CED pathway in pachytene-stage germ cells is necessary for the formation of small germ cells (<20 μm$^2$ basal surface) that undergo apoptosis. Dying germ cells exhibit an enhanced accumulation of actomyosin regulators at their cortex, which causes them to fully constrict and cellularize.

## Actomyosin-mediated apical constriction promotes germ cell death

In many cases, programmed cell death is accompanied by a loss of cell volume [37]. However, it is not known if apoptotic signals are the cause of cell shrinkage, or if a reduction in cell size can also trigger the activation of proapoptotic pathways. Since we found that germ cells undergo apical constriction and reduce in size before forming apoptotic corpses and since apoptotic corpses are enriched in actomyosin regulators, we tested if altering actomyosin contractility in the germline affects the rate of physiological germ cell death.

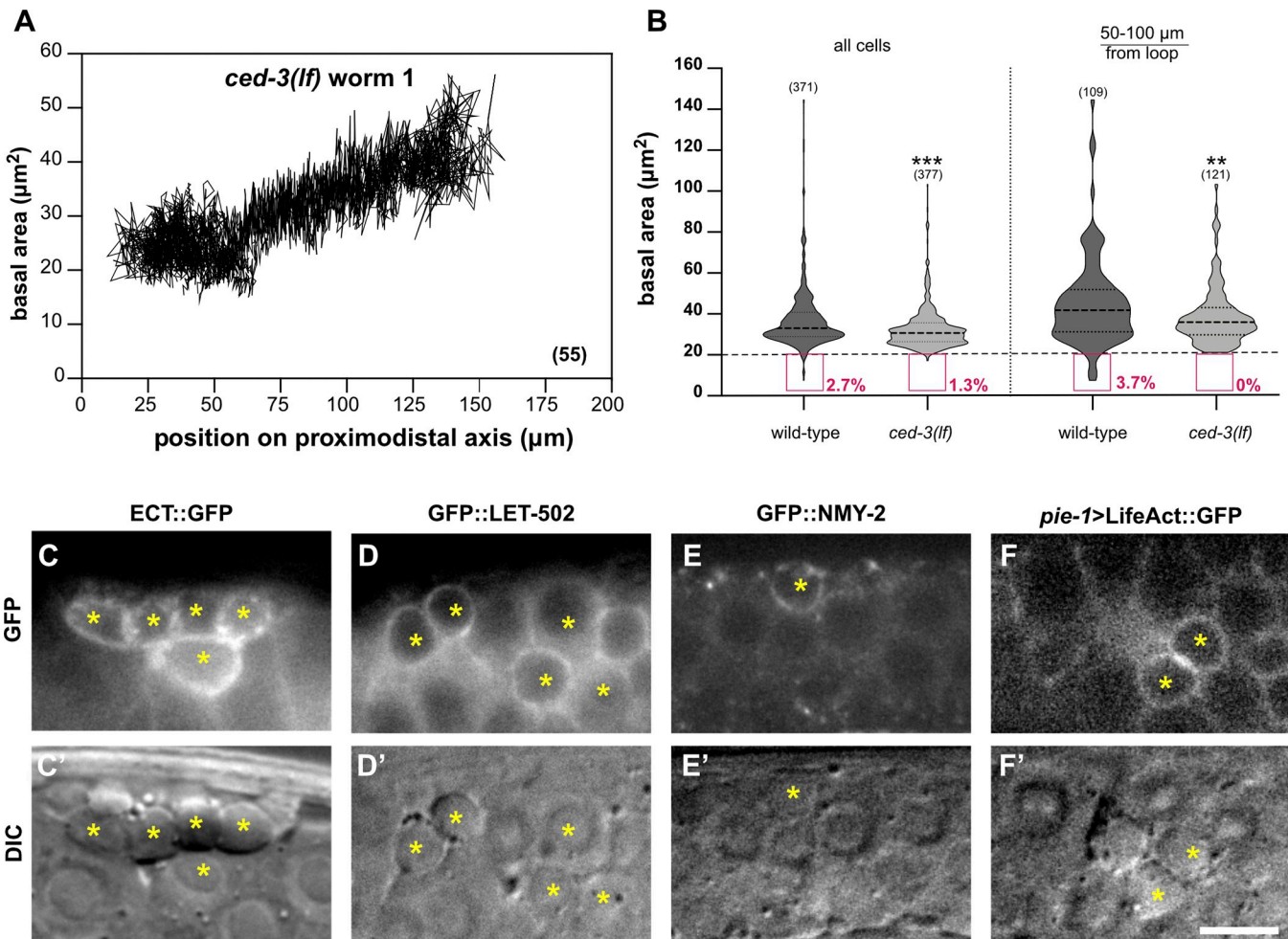

**Fig 2. CED-3 caspase reduces germ cell size.** (**A**) Basal areas of all cells tracked in a *ced-3(lf)* mutant plotted against their relative positions along the distal to proximal axis. Note the absence of small germ cells observed in the wild-type traces in Fig 1F. See S2A and S2B Fig for the traces of 2 additional *ced-3(lf)* animals. (**B**) Violin plots of the basal cell area measured in 1-day-old wild-type and *ced-3(lf)* mutants (8 animals each) in the 0–150 μm region (left) or the mid to late pachytene region (right). The red boxes outline the small cells (area < 20 μm$^2$) and their frequencies. (**C**) Cortical enrichment of the endogenous ECT-2:: GFP, (**D**) GFP::LET-502 and (**E**) NMY-2::GFP reporter signals, and (**F**) a germline-specific transgenic *pie-1*>lifeAct::GFP F-actin marker around apoptotic corpses in 1-day-old adults. The bottom panels (**C'**-**F'**) show the corresponding DIC images, and the yellow asterisks indicate apoptotic cell corpses. Dashed lines in the violin plots indicate the median values and the dotted lines the upper and lower quartiles. The numbers in brackets in each graph refer to the total number of cells analyzed. Statistical significances were calculated with unpaired two-tailed *t* tests. ** indicates *p* < 0.01 and *** for *p* < 0.001. See S1 Data for the underlying data and results of the statistical analysis. Scale bar in (**F'**) is 10 μm.

We first examined if an increase in actomyosin contractility is sufficient to reduce germ cell size and promote germ cell death. To test this, we used a gain-of-function (*gf*) mutation in *ect-2(zh8gf)*, which encodes a guanine–nucleotide exchange factor that activates the RHO-1 small GTPase [23], combined either with the SYN-4::GFP membrane marker to observe germ cell size or with a CED-1::GFP reporter to score germ cell apoptosis [14,38]. CED-1 is a type 1 transmembrane receptor expressed in the sheath cells of the somatic gonad that recognizes and clusters around dying cells before they are engulfed [39]. *ect-2(gf)* mutants contained many very small germ cells from the mid to late pachytene region onwards (magenta dots in **Fig 3B**), interspersed with a few large cells (blue dots) as well as regions devoid of nuclei (green arrowheads; see **S3C and S3D Fig** for full gonad views). Furthermore, many of the

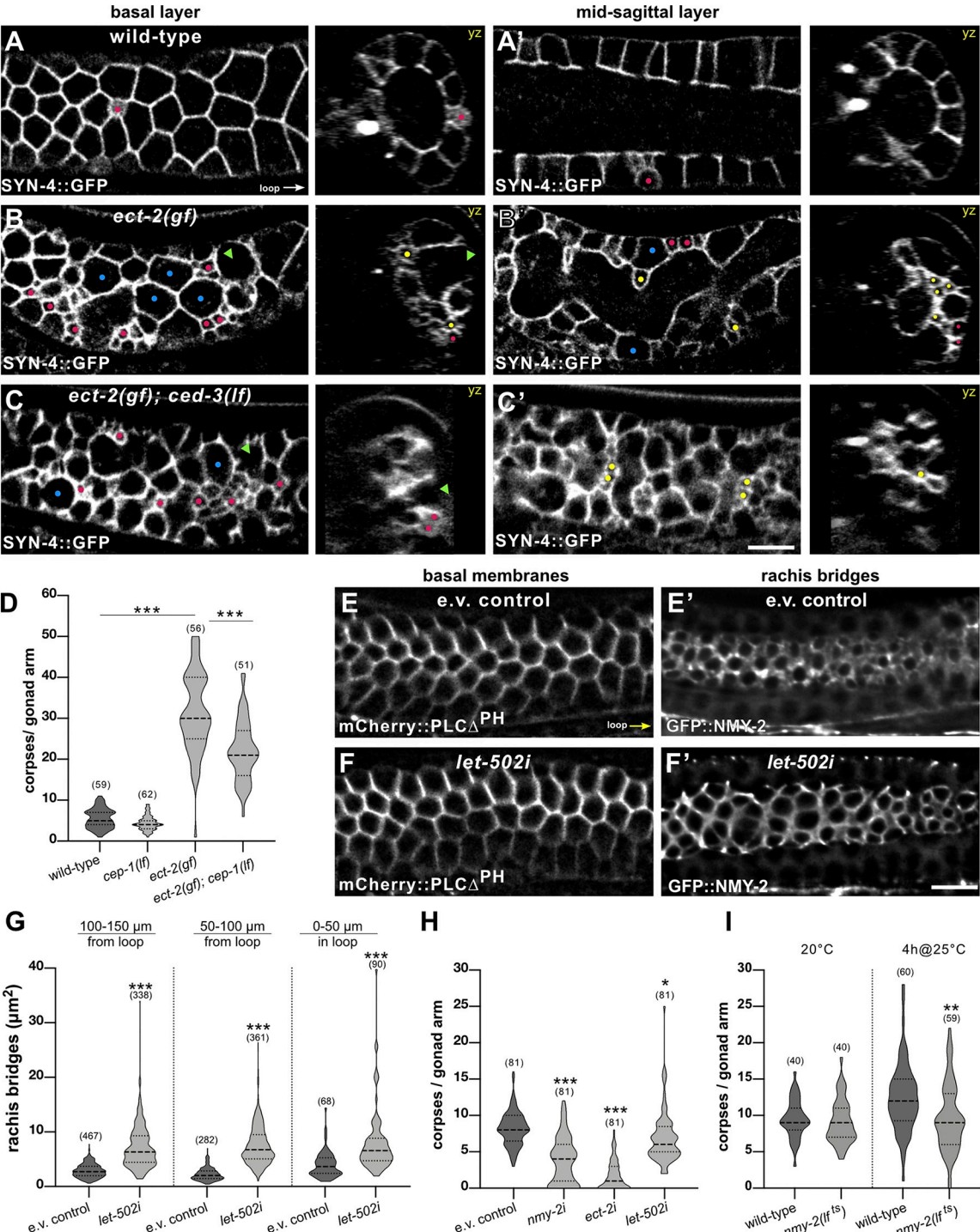

**Fig 3. Actomyosin-mediated apical constriction promotes germ cell death.** Germ cell shapes visualized with the SYN-4::GFP plasma membrane marker in (**A**) wild-type, (**B**) *ect-2(gf)* single, and (**C**) *ect-2(gf); ced-3(lf)* double mutants. For each genotype, optical sections of the basal surface (**A-C**) and the mid-sagittal layers (**A'-C'**) are shown along with the corresponding yz-projections. Magenta dots highlight some of the small cells localized on the surface, yellow dots some of the small cells inside the rachis, blue dots some large cells, and green arrowheads regions devoid of cells. (**D**) Violin plot showing the number of CED-1::GFP-positive apoptotic germ cells per gonad arm in 1-day-old adults of the indicated genotypes. (**E**) Basal germ cell surfaces visualized with the mCherry::PLCΔ^PH membrane marker and (**E'**) the apical rachis bridges outlined with the NMY-2::GFP marker in animals treated with the empty vector (e.v.) as negative control and (**F**, **F'**) after global *let-502 rock* RNAi from the L1 stage on until adulthood. (**G**) Violin plot showing the size of the rachis bridges in the 3 indicated gonad regions in e.v. controls and after global *let-502* RNAi for 72 hours in 1-day-old adults, quantified

as illustrated in S3H Fig. Twelve animals were analyzed for e.v. controls and 13 animals for *let-502i*. (**H**) CED-1::GFP-positive apoptotic germ cells per gonad arm in 1-day-old adults after germline-specific RNAi of the indicated genes. Animals were exposed to dsRNA-producing bacteria from the L4 stage for 24 hours. (**I**) CED-1::GFP-positive cells per gonad arm in 1-day-old wild-type and temperature-sensitive *nmy-2(lf ts)* mutants at the permissive temperature (20°C, left) and after a 4-hour up-shift to the restrictive temperature (4h@25°C, right). Dashed lines in the violin plots indicate the median values and the dotted lines the upper and lower quartiles. For the rachis bridges, the numbers in brackets refer to the number of cells analyzed, and for the CED-1::GFP reporter, the number of animals scored. Statistical analysis was done by one-way ANOVA followed by a Tukey's test for multiple comparisons for Fig 3D, 3H, and 3I or with an unpaired two-tailed *t* test for Fig 3G. * indicates $p < 0.05$, ** $p < 0.01$, and *** $p < 0.001$. See S1 Data for the underlying data and results of the statistical analysis. Scale bars are 10 μm.

small germ cells were found in the central rachis region, which normally contains only cytoplasm (yellow dots in the mid-sagittal layer and the yz-views in **Fig 3B**).

We next tested if the formation of small germ cells in *ect-2(gf)* mutants requires *ced-3 caspase* activity. Small germ cells continued to accumulate in *ect-2(gf); ced-3(lf)* double mutants, indicating that *ect-2* controls germ cell size independently of the CED pathway (**Figs 3C**, **S3E**, and **S3F**). Interestingly, both in *ect-2(gf)* single and *ect-2(gf); ced-3(lf)* double mutants, small germ cells were only present from the mid to late pachytene region on towards the proximal region (**S3C–S3F Fig**). Thus, activation of the ECT/RHO/ROCK pathway alone is not sufficient to reduce germ cell size in more distal regions, as apical constriction may depend on another signal received by germ cells progressing through pachytene.

*ect-2(gf)* mutants exhibited a strong increase in the number of germ cell corpses detected by the CED-1::GFP reporter (**Fig 3D**). To determine if the elevated number of apoptotic germ cells in *ect-2(gf)* mutants was due to DNA damage–induced or physiological cell death, we introduced a *lf* mutation in the p53 homolog *cep-1* [40]. *cep-1(gk138lf)* mutants do not show an increase in germ cell death after DNA damage, while physiological germ cell death occurs normally [30]. *ect-2(gf); cep-1(lf)* double mutants exhibited a significant increase in the number of apoptotic germ cells compared to *cep-1(lf)* single mutants and only a slight reduction relative to *ect-2(gf)* single mutants (**Fig 3D**). Besides inducing actomyosin contractility, ECT-2 also regulates the disassembly of the synaptonemal complexes in pachytene-stage germ cells [41]. Therefore, the increased germ cell death observed after hyperactivation of ECT-2 could be due to chromosome pairing defects. To test this possibility, we examined the effect of a *lf* mutation in *pch-2*, which encodes a component of a meiotic checkpoint that activates apoptosis if unsynapsed chromosomes are detected [42]. *ect-2(gf) pch-2(tm1458)* mutants showed a significant increase in germ cell death compared to *pch-2(tm1458)* single mutant and no decrease relative to *ect-2(gf)* single mutants (**S3G Fig**). By contrast, very few apoptotic corpses were detected in *ect-2(gf); ced-3(lf)* double mutants (**S3H Fig**). Thus, the increased number of apoptotic corpses observed in *ect-2(gf)* mutants is predominantly caused by enhanced physiological germ cell death.

Binucleate germ cells that form in the transition zone and early pachytene region are selectively eliminated by physiological germ cell death in the pachytene region [33]. To examine if the increased germ cell death observed in *ect-2(gf)* mutants is caused by an elevated number of binucleate cells, we counted the numbers of binucleate cells in DAPI-stained gonads dissected from *ect-2(gf)* animals carrying SYN-4::GFP membrane marker (**S3J Fig** and **extended methods** in **S1 File**). The number of binucleate cells in the gonads of *ect-2(gf)* mutants was slightly reduced rather than increased when compared to the wild-type control (**S3J' Fig**). Thus, the enhanced apoptosis caused by hyperactivation of the ECT/RHO/ROCK pathway is not due to an excess formation of binucleate cells.

We then reduced the activities of different actomyosin regulators and scored germ cell death as well as apical germ cell constriction by measuring the size of the apical NMY-2::GFP rings on the rachis bridges. Global RNAi interference (RNAi) against the RHO-dependent

kinase *let-502 rock* [24,25] from the L1 stage onwards increased the sizes of the rachis bridges in pachytene-stage germ cells, as reported previously [19], without perturbing the cellular integrity as examined with the mCherry::PLCΔ$^{PH}$ plasma membrane marker (**Fig 3E–3G;** see **S3I Fig** and **extended methods** in **S1 File** for the quantification of rachis bridge areas). By contrast, constitutive RNAi against the *rho gef ect-2* or the nonmuscle myosin *nmy-2* caused a severe disruption of gonad morphology such that the apical actomyosin corset and lateral germ cell membranes collapsed, precluding the quantification of germ cell death and apical constriction.

We therefore performed germline-specific RNAi using a strain expressing the Argonaute protein RDE-1 exclusively in the germ cells [43] and the CED-1::GFP marker to score germ cell death. Germline-specific RNAi against *nmy-2*, *ect-2*, or *let-502* from the L4 stage decreased the number of CED-1::GFP-positive cells in 1-day-old adults when compared to empty vector–treated control animals (**Fig 3H**). It should be noted that the stronger decrease in CED-1::GFP-positive germ cells after *ect-2* RNAi, but not *nmy-2* or *let-502* RNAi, may in part be due to the collapse of the plasma membranes since the gonad architecture appeared disorganized and clumps of nuclei formed in around 70% of *ect-2* RNAi-treated animals. Also, a transient 4-hour-long inactivation of NMY-2 using the temperature-sensitive *nmy-2(ne1490ts)* allele [44] caused a reduction in the number of CED-1::GFP-positive cells at the restrictive temperature (**Fig 3I**).

Taken together, these data indicate that a decrease in actomyosin contractility increases the size of the apical rachis bridges and reduces germ cell apoptosis, while hyperactivation of the ECT/RHO/ROCK pathway is sufficient to increase the rate of germ cell apoptosis. Thus, apical germ cell constriction promotes physiological germ cell death.

## RAS/MAPK signaling promotes physiological germ cell death

Previous studies have shown that the RAS/MAPK pathway is an essential regulator of physiological and damage-induced germ cell death [11,30,45]. An increase in MAPK activity at the late pachytene stage results in the formation of more but smaller oocytes and an elevated number of apoptotic corpses, while reduced MAPK activity delays pachytene exit and reduces the number of oocytes in the proximal gonads [11,31,46–48]. Different effects on germ cell apoptosis caused by reducing MAPK activity have been reported. Some studies observed decreased apoptosis [11,30,46], while others observed increased apoptosis [48,49] after global inhibition of the MAP kinase MPK-1 using the temperature-sensitive *mpk-1(ga111)* allele [50].

Consistent with the previous reports, we observed an elevated number of germ cell corpses after transient hyperactivation of the RAS/MAPK pathway by growing temperature-sensitive *let-60(ga89$^{ts}$) ras gf* mutants (abbreviated *let-60(gf$^{ts}$)*) for 4 hours at the restrictive temperature of 25°C (**Fig 4A**) [51]. The increase in germ cell apoptosis in *let-60(gf$^{ts}$)* animals was independent of *cep-1* and *pch-2*, indicating that elevated RAS/MAPK signaling stimulates physiological germ cell death without activating a DNA damage response or meiotic checkpoint (**Figs 4A and S4B**). The number of binucleate germ cells in *let-60(gf$^{ts}$)* animals was reduced rather than increased (**S3J' Fig**). Thus, the increase in germ cell death caused by enhanced RAS/MAPK signaling is not due to the formation of more binucleate germ cells. On the other hand, germ cell death in *let-60(gf$^{ts}$)* mutants completely depended on *ced-4* activity (**S4A Fig**), indicating that RAS/MAPK signaling promotes physiological germ cell death via the canonical CED pathway [7,11].

Furthermore, *lf* mutations in *gap-3* and *gap-1*, which encode GTPase activating proteins that negatively regulate LET-60 RAS activity [52], also enhanced germ cell death (**S4F Fig**).

To investigate the effects of reduced RAS/MAPK signaling on germ cell death, we first performed tissue-specific RNAi knock-down of the MAP kinase *mpk-1*. Global *mpk-1* RNAi in

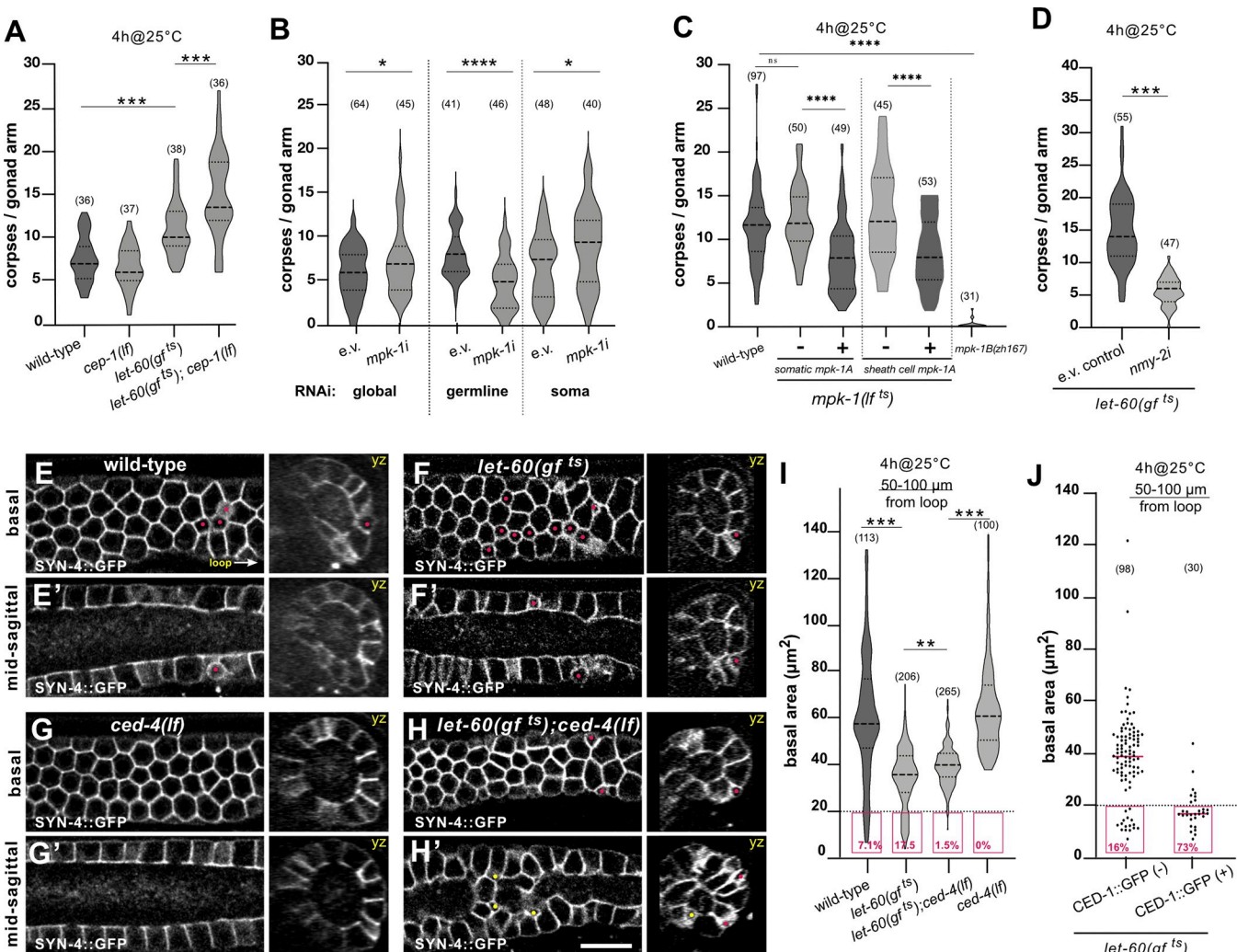

**Fig 4. RAS/MAPK signaling promotes apoptosis.** (**A**) Violin plot showing the number of CED-1::GFP-positive apoptotic germ cells per gonad arm in 1-day-old adults of the indicated genotypes grown at 20°C for 68 hours and subsequently up-shift for 4 hours to the restrictive temperature (4h@25°C). (**B**) mCherry::ΔPH-positive apoptotic germ cells in 1-day-old adults after global, germline-specific and soma-specific *mpk-1* RNAi. Here, we used the *zhIs198[mCherry::ΔPH]* corpse marker because it shows less interference to *mpk-1* RNAi treatment than the *ced-1::gfp* corpse marker. (**C**) CED-1::GFP-positive apoptotic germ cells in wild-type, temperature-sensitive *mpk-1(lf ts)* mutants without and with the zh*Ex676[somatic mpk-1a]* transgene expressing the *mpk-1a* isoform in the soma, without and with the sheath cell–specific *narEx35[Pckb-3>mpk-1a]* transgene, and in germline-specific *mpk-1b(zh164lf)* mutants. Twenty hours post L4, the animals were grown for 4 hours at the restrictive temperature of 25°C (4h@25°C). The dashed lines indicate that the sheath cell–specific rescue experiment was scored on a different day. Controls grown continuously at 20°C are shown in S4C Fig. (**D**) CED-1::GFP-positive apoptotic germ cells in 1-day-old adult *let-60 (gf ts)* mutant treated with empty vector or *nmy-2i*. Animals were RNAi treated for 24 hours from the L4 stage on and up-shifted to 25°C for the last 4 hours before scoring the corpses (for 24 hours up-shift at 25°C, see S4D Fig). (**E**) Germ cell shapes visualized with the SYN-4::GFP plasma membrane markers in wild-type, (**F**) in *let-60(gf ts)*, (**G**) in *ced-4(lf)* single, and in (**H**) *let-60(gf ts); ced-4(lf)* double mutant. For each genotype, optical sections of the basal surface (**E**-**H**) and the mid-sagittal layers (**E'**-**H'**) are shown along with the corresponding yz-projections. Magenta dots highlight small cells localized on the surface and yellow dots small cells inside the rachis. (**I**) Violin plot of the basal cell areas in the late pachytene region (50–100 μm from the loop) measured using MorphographX in 1-day-old adults of the indicated genotypes. Twenty hours post L4, the animals were grown for 4 hours at the restrictive temperature of 25°C (4h@25°C). The magenta boxes outline the small cells (area < 20 μm$^2$), and their frequencies are shown as percentage values. Results of the size measurements in the other gonad regions are shown in S4E Fig. Ten animals were analyzed for wild type, 15 animals for *let-60(gf ts)*, 16 animals for *let-60(gf ts); ced-4(lf)*, and 9 animals for *ced-4(lf)*. (**J**) Basal cell areas in the late pachytene region (50–100 μm region) of CED-1::GFP-negative cells versus CED-1::GFP-positive cells in 1-day-old adults of *let-60(gf ts)* grown at 20°C for 68 hours and incubated for 4 hours at the restrictive temperature of 25°C. Fifteen *let-60(gf ts)* animals were analyzed. Dashed lines in the violin plots indicate the median values and the dotted lines the upper and lower quartiles. For the measurements of the basal areas, the numbers in brackets refer to the numbers of cells analyzed, and for the CED-1::GFP reporter to the numbers of animals scored. Statistical analysis was done using one-way ANOVA followed by a Tukey's test for multiple comparisons or with an unpaired two-tailed *t* test. ** indicates *p* < 0.01 and *** *p* < 0.001. See S1 Data for the underlying data and results of the statistical analysis. Scale bars are 10 μm.

the soma and germline resulted in a very slight increase in the number of apoptotic corpses (**Fig 4B**). Germline-specific *mpk-1* RNAi, on the other hand, reduced the number of apoptotic corpses, whereas soma-specific *mpk-1* RNAi caused an increase in corpse number (**Fig 4B**).

Similar effects were observed using the temperature-sensitive *mpk-1(ga111 ts)* reduction-of-function allele (abbreviated *mpk-1(lf ts)*) [50]. A transient reduction of MPK-1 activity in the soma and germline induced by growing *mpk-1(lf ts)* mutants for 4 hours at the restrictive temperature of 25˚C did not result in a significant change in the number of germ cell corpses (**Fig 4C**). However, *mpk-1(lf ts)* mutants expressing the *mpk-1a* isoform from an extrachromosomal array in the entire soma or in the sheath cells of the somatic gonad [53] contained a significantly reduced number of germ cell corpses at the restrictive temperature (**Figs 4C** and **S4C**).

To further distinguish between the activities of the somatic *mpk-1a* and the germline-specific *mpk-1b* isoforms, we generated the germline-specific *mpk-1b(zh164)* allele (abbreviated *mpk-1b(lf))* by introducing a stop codon in the first exon of the *mpk-1b* isoform (W4Stop; see **Materials and methods**). Homozygous *mpk-1b(lf)* animals exhibited a fully penetrant sterile phenotype due to disrupted gonad architecture and absent oogenesis in 1-day-old adults, but no obvious defects in the soma were observed. Germline-specific *mpk-1b(lf)* mutants contained only few germ cell corpses (**Figs 4C** and **S4C**).

We thus conclude that the somatic *mpk-1a* and the germline-specific *mpk-1b* isoforms have opposing effects on germ cell death. The somatic *mpk-1a* isoform in the gonadal sheath cells suppresses corpse formation, possibly by inducing the clearing of apoptotic corpses, whereas the germline-specific *mpk-1b* isoform promotes physiological germ cell death. Hence, if MAPK signaling is simultaneously reduced in the soma and germline, a decreased rate of germ cell apoptosis may be compensated for by reduced MPK-1A activity in the gonadal sheath cells.

## Hyperactivation of RAS/MAPK signaling decreases basal germ cell area

Hyperactivation of the RAS/MAPK pathway or enhanced RHO/ROCK signaling both resulted in an elevated rate of physiological germ cell death. These observations raised the possibility that enhancing RAS/MAPK signaling might also cause a reduction in germ cell size, which, in turn, would promote germ cell death. To test this hypothesis, we performed RNAi against the nonmuscle myosin NMY-2 in the temperature-sensitive *let-60(gf ts)* background and observed the CED-1::GFP marker to score germ cell apoptosis and the mCherry::PLCΔ$^{PH}$ membrane marker to assess plasma membrane integrity. RNAi against *nmy-2* was carried out for 24 hours from the L4 stage on, combined with a hyperactivation of *let-60 ras* signaling at 25˚C for the last 4 hours, or with a simultaneous 24-hour-long hyperactivation of *let-60 ras* from the L4 stage on. Both treatments decreased the number of CED-1::GFP-positive germ cells (**Figs 4D** and **S4D**). The increase in physiological germ cell death observed after hyperactivation of the RAS/MAPK pathway therefore depends, at least in part, on the activity of the actomyosin network.

We next examined germ cell size by observing the SYN-4::GFP plasma membrane marker in temperature-sensitive *let-60(gf ts)* mutants grown for 4 hours at the restrictive temperature. Hyperactivation of the RAS/MAPK pathway resulted in the formation of more small germ cells (basal area below 20 μm$^2$), especially at the late pachytene stage 50 to 100 μm before the loop (red dots in **Fig 4E and 4F** and magenta boxes in **4I**). Also, the average size of all germ cells combined was reduced in *let-60(gf ts)* mutants (**Figs 4I** and **S4E**).

The effect of the RAS/MAPK pathway on germ cell size may be indirect. For example, RAS/MAPK signaling could directly activate the proapoptotic CED pathway, which would cause

apoptotic germ cells to shrink. Alternatively, RAS/MAPK signaling may regulate germ cell size through the actomyosin pathway independently of the CED pathway.

To distinguish between these 2 possibilities, we quantified germ cell sizes in *let-60(gf$^{ts}$); ced-4(lf)* double mutants, as well as in *ced-4(lf)* single mutants. Small germ cells (basal area below 20 μm$^2$) accumulated in the pachytene region of *let-60(gf$^{ts}$); ced-4(lf)* double mutants mostly inside the rachis (**Fig 4H and 4I**), while *ced-4(lf)* single mutants did not contain any small germ cells (**Fig 4G and 4I**), like *ced-3(lf)* mutants (**Fig 2A and 2B**). Since an accurate quantification of germ cells inside the rachis was not possible using our MorphographX pipeline, the fraction of small cells shown in **Fig 4I** for *let-60(gf$^{ts}$); ced-4(lf)* double mutants is probably an underestimate, as it only includes small cells at the basal surface. Moreover, in the late pachytene region (50 to 100 μm before the loop), the average germ cell size in *let-60(gf$^{ts}$); ced-4(lf)* double mutants was significantly smaller than in *ced-4(lf)* single mutants and more similar to *let-60(gf$^{ts}$)* single mutants, except for the reduced fraction of very small cells (**Figs 4I** and **S4E**).

To directly correlate germ cell size and apoptosis in *let-60(gf$^{ts}$)* mutants, we compared the sizes of CED-1::GFP-positive cells to their CED-1::GFP-negative neighbors in the late pachytene region (50 to 100 μm) using the mCherry::PLCΔ$^{PH}$ membrane marker (**Fig 4J** and **extended methods** in **S1 File**). Small germ cells (basal area below 20 μm$^2$) were more likely to be CED-1::GFP positive (73%) than larger cells. It should be noted that also a population of CED-1::GFP-negative small cells (16%) was observed. Since the germ cell tracking experiments (**Fig 1**) indicated that germ cells with a basal area below 20 μm$^2$ have a 58% probability of forming an apoptotic corpse within 1 hour, the population of CED-1::GFP-negative small cells may contain preapoptotic cells that have not yet been engulfed.

Taken together, our data indicate that RAS/MAPK signaling decreases germ cell size independently of the CED pathway. The small germ cells generated in the wild type or after hyperactivation of the RAS/MAPK pathway are more likely to be engulfed than their larger neighbors.

## RAS/MAPK signaling induces apical germ cell constriction

We have previously reported that activation of the RAS/MAPK pathway by the EGF receptor induces apical constriction of the primary vulval precursor cells during morphogenesis of the hermaphrodite vulva [54]. This finding raised the possibility that RAS/MAPK signaling could also induce apical constriction at the rachis bridges of germ cells in the pachytene region to regulate germ cell size and death.

To investigate this hypothesis, we measured the sizes of the rachis bridges after hyperactivation or inhibition of RAS/MAPK signaling, by combining the temperature-sensitive *let-60(gf$^{ts}$)* and *mpk-1(lf$^{ts}$)* alleles with the endogenous NMY-2::GFP nonmuscle myosin marker, which outlines the rachis bridges (**Fig 5A**). In 1-day-old adult *let-60(gf$^{ts}$)* animals grown for 4 hours at the restrictive temperature, the average area of the rachis bridges was decreased (**Figs 5A–5C, S5A,** and **S5B**). In many cases, the NMY-2::GFP signal accumulated in single bright spots, which could not be quantified, on the apical side facing the rachis (yellow arrows in **S5A and S5B Fig**), suggesting that the apical rachis bridges in those cells had already fully constricted. Conversely, in young adult *mpk-1(lf$^{ts}$)* animals grown for 4 or 6 hours at the restrictive temperature, the average area of the rachis bridges was increased (**Figs 5D–5F** and **S5C–S5E**).

To study the effects of altered RAS/MAPK signaling on gonad architecture, we compared the heights of the germ cells as well as the rachis diameter in the pachytene region (50 to 150 μm from the loop) of wild-type, *let-60(gf$^{ts}$)*, and *mpk-1(lf$^{ts}$)* mutants. To measure the average rachis diameters, the total areas of the regions shown in **Fig 5J'–5L'** were measured in mid-sagittal sections and divided by their length (100 μm). In 1-day-old *let-60(gf$^{ts}$)* mutants

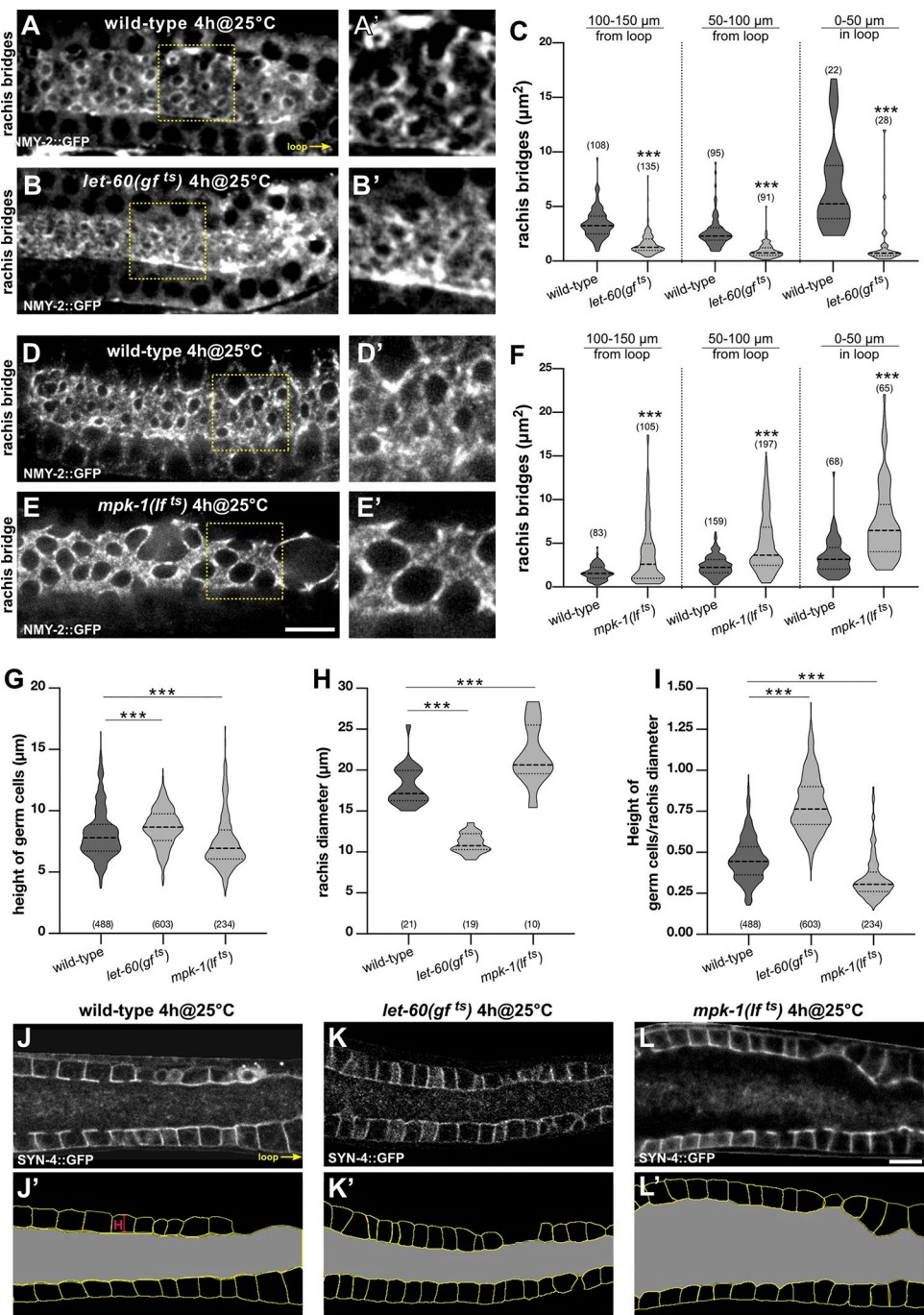

**Fig 5. RAS/MAPK signaling promotes apical germ cell constriction.** (**A**, **B**) Rachis bridges outlined by the NMY-2::GFP reporter in 1-day-old wild-type and *let-60(gf^ts)* mutant animals grown for 4 hours at the restrictive temperature of 25˚C. (**A'** and **B'**) Two-fold magnifications of the regions outlined by the dashed yellow boxes in (**A**) and (**B**). Additional examples of the rachis bridges in wild-type and *let-60(gf^ts)* mutants are shown in S5A and S5B Fig. (**C**) Violin plots showing the size of the rachis bridges in wild-type and *let-60(gf^ts)* mutant animals in the 3 indicated gonad regions. Five animals were analyzed for wild type and 6 animals for *let-60(gf^ts)*. (**D**, **E**) Rachis bridges in wild-type and *mpk-1(lf^ts)* mutant animals grown for 4 hours at the restrictive temperature (4h@25°C) at the L4 stage. (**D'** and **E'**) Two-fold magnifications of the regions outlined by the dashed yellow boxes in (**D**) and (**E**). (**F**) Violin plots showing the size of the rachis bridges in wild-type and *mpk-1(lf^ts)* mutant animals in the 3 indicated gonad regions. Nine animals were analyzed for wild type and 8 animals for *mpk-1(lf^ts)*. The quantification of the rachis bridges after a 6-hour inactivation of MPK-1 is shown in S5C–S5E Fig. (**G**) Violin plots showing the height of pachytene germ cells (lateral membranes 50 to 150 μm from the loop) in wild-type, *let-60(gf^ts)*, and *mpk-1(lf^ts)* mutants, grown at 20˚C for

68 hours and incubated for the last 4 hours at 25˚C. Twenty-one animals were analyzed for wild type, 19 for *let-60(gf ts)*, and 10 for *mpk-1(lf ts)*. (**H**) Violin plots show the average diameter of the rachis in the same animals as in (**G**). (**I**) Violin plots showing the ratios of germ cell height to rachis diameter. (**J-L**) Mid-sagittal sections of a wild-type, a *let-60 (gf ts)*, and an *mpk-1(lf ts)* mutant were used to measure cell height and rachis diameter shown in (**G-I**). (**J'-L'**) Masks generated from the animals shown in (**J-L**) using the cellpose algorithm [55]. Cell borders are shown in yellow and the rachis area is in grey. Dashed lines in the violin plots indicate the median values and the dotted lines the upper and lower quartiles. For the measurements of the rachis bridges and cell heights, the numbers in brackets refer to the number of cells analyzed, and for the rachis diameter to the number of animals scored. Statistical analysis was done using one-way ANOVA followed by a Tukey's test for multiple comparisons or with an unpaired two-tailed *t* test. ** indicates $p < 0.01$ and *** $p < 0.001$. See S1 Data for the underlying data and results of the statistical analysis. Scale bars are 10 μm.

incubated for 4 hours at 25˚C, the average rachis diameter was decreased by 39%, while the average height of the cells was increased by 6.6% relative to the wild type. Conversely, in *mpk-1 (lf ts)* mutants incubated at 25˚C for 4 hours, the average rachis area was expanded by 21%, while the average height of the cells was decreased by 8.6% (**Fig 5G–5I**).

Together, these results suggest that RAS/MAPK signaling promotes actomyosin constriction at the apical rachis bridges. In addition, elevated RAS/MAPK signaling results in a decreased diameter of the central rachis and elongation of the pachytene germ cells along their apicobasal axis.

## RAS/MAPK signaling is necessary for NMY-2 myosin enrichment at the rachis bridges

To investigate the interactions between the ECT/RHO/ROCK and RAS/MAPK pathways, we combined the germline-specific *mpk-1b(lf)* or the temperature-sensitive *mpk-1(lf ts)* alleles with the *ect-2(gf)* mutation and scored germ cell apoptosis using the CED-1::GFP marker. Especially, the germline-specific *mpk-1b(lf)* allele excludes possible interference caused by loss of the somatic MPK-1A isoform. Animals were scored 24 hours after the L4 stage at 20˚C or for the *mpk-1(lf ts)* allele after incubation at 25˚C for the final 4 hours. *ect-2(gf); mpk-1b(lf)* double mutants displayed a strong reduction in CED-1::GFP-positive cells compared to *ect-2 (gf)* single mutants (**Fig 6A**, left). Also in *ect-2(gf); mpk-1(lf ts)* double mutants, the number of CED-1::GFP-positive cells decreased to the level of *mpk-1(lf ts)* single mutants (**Fig 6A**, right). Therefore, MPK-1 acts either downstream of ECT-2, or RAS/MAPK signaling promotes germ cell death in parallel with the ECT/RHO/ROCK pathway through a different mechanism.

To examine if MAPK activity is regulated by the ECT/RHO/ROCK pathway, we quantified the levels of the activated, diphosphorylated 50.6 kD MPK-1B isoform (P-MPK-1B) by western blot analysis using a diphospho-specific ERK antibody [31]. Analysis of whole-animal extracts of 1-day-old adult *ect-2(gf)* and *let-502(lf ts)* animals grown for 4 hours at the restrictive temperature of 25˚C revealed no significant changes in P-MPK-1B levels (**Figs 6B** and **S6A– S6A"**). Thus, the RAS/MAPK and ECT/RHO/ROCK signaling pathways likely act through distinct mechanisms to induce apical germ cell constriction and death.

To investigate how RAS/MAPK signaling controls germ cell constriction, we examined the subcellular distribution of NMY-2::GFP myosin. In the late pachytene region (50 to 100 μm from the loop) of 1-day-old adult wild-type animals, NMY-2::GFP was enriched in punctate structures around the rachis bridges (**Fig 6C**; see **S6B Fig** for intensity profiles and additional examples). In *mpk-1b(lf)* mutants, on the other hand, NMY-2::GFP enrichment at the rachis bridges was reduced, and the NMY-2::GFP puncta were evenly distributed on the apical cortex between the rachis bridges (**Figs 6D** and **S6C**). Since the diameter of the rachis bridges was enlarged in *mpk-1b(lf)* mutants, and, hence, the area of the apical cortex between the rachis bridges reduced, *mpk-1b(lf)* likely resulted in a higher local NMY-2::GFP concentration on the

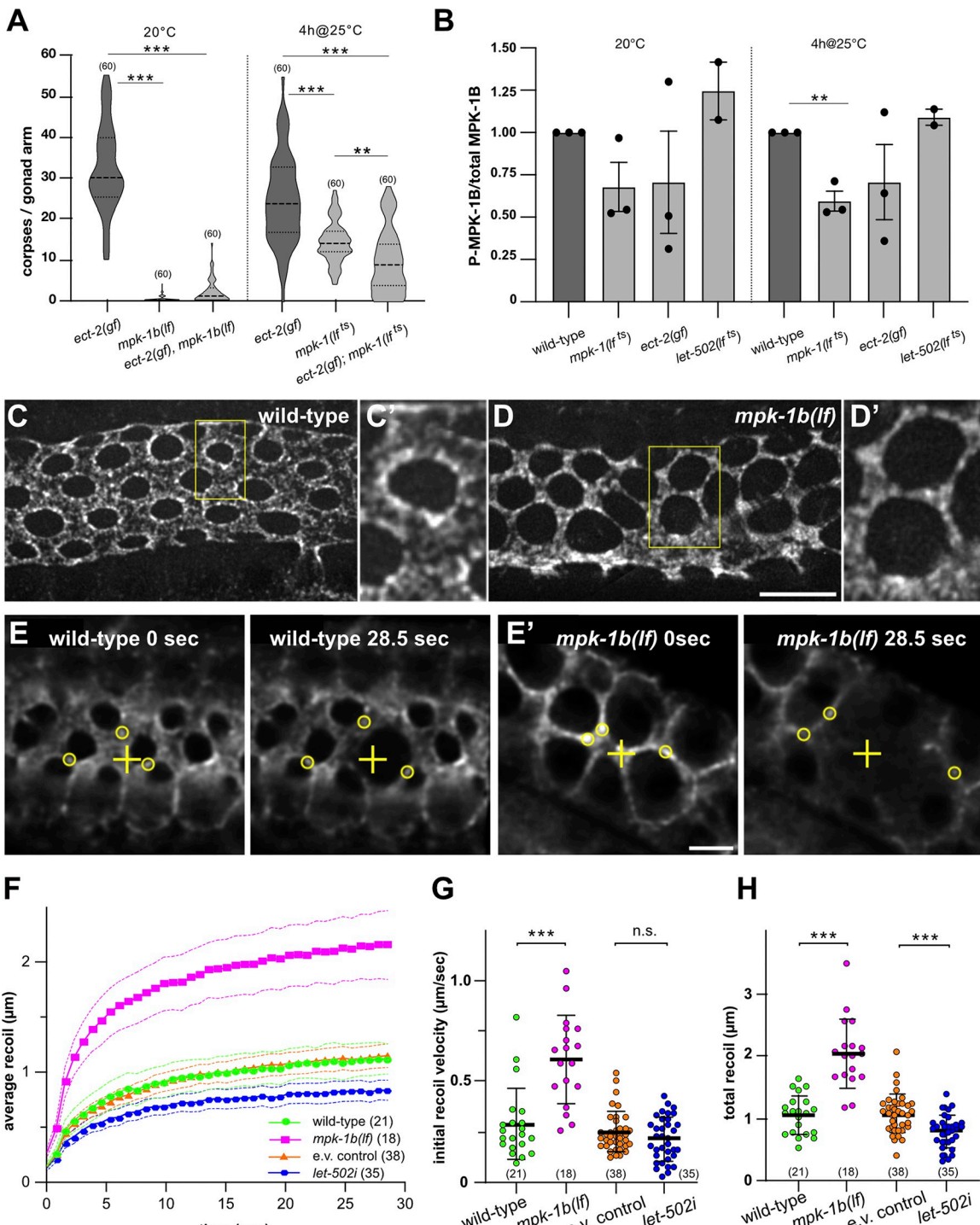

**Fig 6. RAS/MAPK signaling controls NMY-2 myosin localization at the rachis bridges.** (**A**) Violin plot showing the number of CED-1::GFP-positive apoptotic germ cells per gonad arm in 1-day-old adults of the indicated genotypes, grown either at 20°C (left) or up-shifted to 25°C for the last 4 hours (right). (**B**) Quantification of diphosphorylated MPK-1B (P-MPK-1B) and total MPK-1B protein levels on western blots of total extracts from 1-day-old adults of the indicated genotypes, incubated either at 20°C or for the final 4 hours at 25°C. The average intensity ratios ± SEM of the P-MPK-1B to MPK-1B signals, each normalized to the tubulin loading controls and relative to the wild-type control values obtained in each experiment are shown. Quantification of 3 biological replicates as described in Materials and methods, except for 2 *let-502* replicates. Individual blots are shown in **S6A–S6A" Fig**. (**C**) NMY-2::GFP localization in wild-type and (**D**) *mpk-1b(lf)* mutants in the late pachytene region (50–100 μm from the loop) 66 to 68 hours post-L1 arrest. Maximum intensity projections of deconvolved z-stacks spanning the apical surface are shown. The yellow boxes outline the regions shown at

2-fold magnification in (**C'**) and (**D'**). NMY-2::GFP intensity profiles across the rachis bridges and additional examples are shown in **S6B and S6C**. (**E**) Recoil after point incision in wild-type and (**E'**) *mpk-1b(lf)* 1-day-old adults (66 to 68 hours after L1 arrest). The left frames (0 seconds) show the apical surface before and the right frames 28.5 seconds after the incision. The yellow crosses indicate the incision point and the circles mark NMY-2::GFP foci used to track the mean radial displacement from the incision points. See **S3** and **S4 Movies** for all time points of the examples shown, and **S1 Data** for the individual measurements of all animals scored. (**F**) Mean radial displacement (recoil) plotted against time after incision. The symbols indicate the average recoil of all animals analyzed and the dashed lines the 95% CI. (**G**) Initial recoil velocities and (**H**) total recoil in the indicated genotypes/conditions were calculated for each animal individually. See Materials and methods for details on the curve fitting and quantification. Dashed horizontal lines in the violin plots in (**A**) indicate the median values and dotted lines the upper and lower quartiles, solid horizontal lines in (**G**) and (**H**) the mean values ± SD. The numbers in brackets refer to the numbers of animals analyzed. Statistical analysis was done using one-way ANOVA followed by a Tukey's test for multiple comparisons (**A, B**) or by unpaired *t* tests (**G, H**). ** indicates $p < 0.01$ and *** $p < 0.001$. See **S1 Data** for the underlying data and results of the statistical analysis. Scale bars in (**D**) and (**E'**) are 10 μm and 5 μm, respectively.

apical cortex. Quantification of the average NMY-2::GFP intensities over the entire apical surface including the rachis bridges indicated that loss of *mpk-1b* did not significantly alter total NMY-2::GFP levels (**S6D and S6D' Fig**). Reduced NMY-2::GFP localization at the rachis bridges was also observed in *mpk-1(lf$^{ts}$)* mutants, although the effect was less pronounced due to the partial inactivation of *mpk-1* (**Figs 5E** and **S5D**).

Thus, RAS/MAPK signaling may be necessary to localize NMY-2 to the rachis bridges. However, it should be noted that the lower levels of NMY-2::GFP at the rachis bridges could also be an indirect consequence of the changed gonad morphology (i.e., the enlarged rachis bridges) in *mpk-1(lf)* mutants.

To test if the altered distribution of NMY-2 in *mpk-1b(lf)* and *mpk-1(lf$^{ts}$)* mutants changed actomyosin contractility, we performed laser ablation experiments to measure the cortical tension, as described [18,56]. Using a pulsed UV laser, we created point incisions on the apical membranes between the rachis bridges, followed the expansion of the incisions by tracking the displacement of individual NMY-2::GFP foci (**Figs 6E, 6F,** and **S6E**), and quantified the initial recoil velocity and total recoil as indicators of the cortical tension [53] (see also Materials and methods). The initial recoil velocity and the total recoil were both increased in *mpk-1b(lf)* mutants grown at 20°C (**Fig 6G and 6H**) and in *mpk-1(lf$^{ts}$)* mutants grown for 4 hours at 25°C (**S6F and S6G Fig**). On the other hand, RNAi of *let-502 rock* resulted in a smaller total recoil, although the initial recoil velocity was not significantly changed (**Fig 6F–6H**).

We conclude that RAS/MAPK signaling is not necessary to induce actomyosin contractility, but rather to control the subcellular distribution of NMY-2 myosin. The increased recoil in *mpk-1* mutants may be due to the changed NMY-2 localization, resulting in a higher local concentration of NMY-2 on the apical cortex. Inhibition of *let-502 rock*, on the other hand, causes a global reduction of actomyosin contractility and a reduced tension on the apical cortex.

## Discussion

Apoptotic cell death is a ubiquitous process that serves to maintain cellular homeostasis in self-renewing organs [2–4]. We have investigated germ cell apoptosis in *C. elegans* hermaphrodites, a physiological process that eliminates around 60% of all germ cells during the pachytene stage of meiotic prophase I. In contrast to the programmed cell death occurring during embryo and larval development in the *C. elegans* soma, physiological germ cell death in adult animals appears to be a stochastic process that eliminates randomly selected cells.

Here, we show that pachytene-stage germ cells smaller than their neighbors are selectively eliminated through apoptosis, probably to make space for and donate their resources to the surviving germ cells entering oogenesis [11,15]. Apical actomyosin constriction at the rachis bridges that connect the cells to a common cytoplasmic reservoir (the rachis) is jointly controlled by the RAS/MAPK and ECT/RHO/ROCK signaling pathways. Enhancing apical germ

cell constriction reduces germ cell size and thereby increases the rate of physiological germ cell death (**Fig 7**).

## The RHO/ROCK pathway promotes physiological germ cell death via apical germ cell constriction

Based on our germ cell tracking data, which demonstrate that germ cells reduce their size 1 to 2 hours before forming apoptotic corpses, we investigated the involvement of the RHO/ROCK pathway in physiological germ cell death. Reducing germ cell size by hyperactivation of the RHO GEF ECT-2 was sufficient to increase germ cell death at the pachytene stage, whereas inhibiting the RHO pathway specifically in germ cells caused a reduction in physiological germ cell death. These data indicated that reduced germ cell size could be a cause and not only a consequence of germ cell death. To exclude the possibility that the reduction in germ cell death observed after inhibiting the ECT/RHO/ROCK pathway was an indirect consequence of an overall loss of cellular integrity, we monitored germ cell shape and chose conditions that only transiently inhibited the ECT/RHO/ROCK pathway without perturbing gonad architecture. Under these conditions, we observed a reduction in apoptotic corpse formation while maintaining germ cell integrity.

The changes in germ cell size and death caused by reducing actomyosin contractility are accompanied by the opening of the rachis bridges. This suggests that germ cell size is regulated by the constriction of the rachis bridges, which regulate the flow of cytoplasm between the germ cells and the rachis. Interestingly, the constriction of the rachis bridges and apoptotic corpse formation only occurred in the mid to late pachytene region, before the surviving germ cells enter the loop region, where they reopen their rachis bridges and grow in size. These observations suggest that germ cells progressing through the pachytene may receive a positional signal that induces apical germ cell constriction. Moreover, the constitutive activation of the ECT/RHO/ROCK pathway through a *gf* mutation in *ect-2* reduced germ cell size only from the mid to late pachytene region onwards. Thus, apical germ cell constriction in the pachytene region may depend on additional signals, which may be transduced by the RAS/MAPK pathway.

## RAS/MAPK signaling induces apical germ cell constriction

The RAS/MAPK pathway regulates oogenesis in various organisms, including *C. elegans*, *Drosophila*, and mammals [47,57–59]. In the *C. elegans* hermaphrodite germline, RAS/MAPK signaling plays an essential role in 4 processes, during mitotic proliferation, pachytene progression, oocyte maturation, and male germ cell fate specification [31,53,60]. The activation of RAS/MAPK signaling by the DAF-2 InsR in germ cells progressing through the pachytene stage is essential for the germ cells to maintain membrane integrity, exit pachytene, and initiate oocyte differentiation [29,31,61].

Previous reports have shown that RAS/MAPK signaling at the pachytene stage is required for physiological as well as DNA damage–induced germ cell death [11,30]. We have confirmed this for the physiological cell death by specific inactivation of the germ cell–specific MPK-1b isoform and by using the hypomorphic *mpk-1(ga111ts)* allele combined with somatic rescue of *mpk-1a*. Moreover, we show that RAS/MAPK signaling reduces germ cell size by inducing apical constriction at the rachis bridges together with the ECT/RHO/ROCK pathway discussed above. The regulation of germ cell size by RAS/MAPK signaling does not require the activation of the proapoptotic CED pathway, indicating that the reduced size of the germ cells in *let-60 (gf)* mutants is not a consequence of enhanced apoptosis.

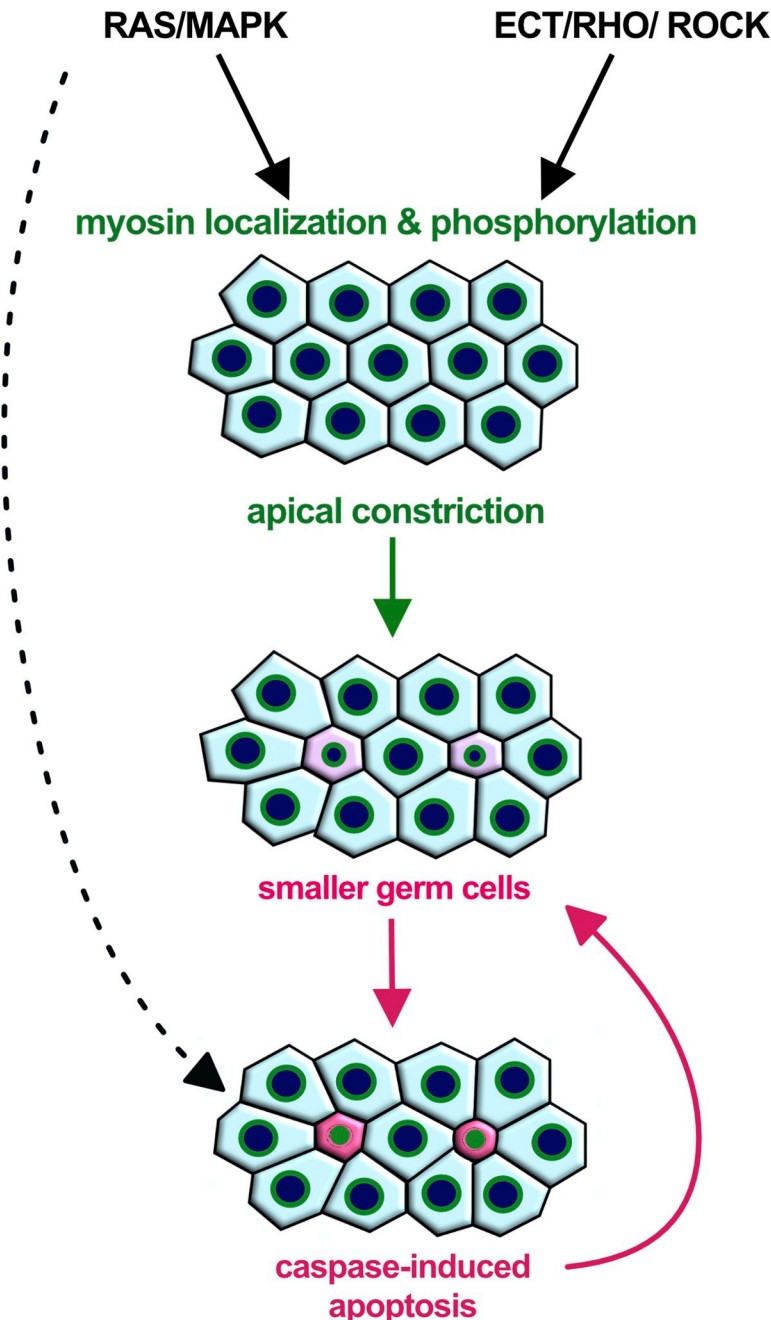

**Fig 7. Apical germ cell constriction promotes physiological germ cell death.** Activation of RAS/MAPK signaling in germ cells at the pachytene stage is necessary for the localization of NMY-2 myosin at the rachis bridges, while ECT-2/RHO/ROCK signaling regulates actomyosin contractility via phosphorylation of the regulatory myosin light chain MLC-4. Actomyosin contractility at the rachis bridges controls germ cell size and enhances stochastic disparities in germ cell sizes (Laplace effect). Smaller germ cells are selectively eliminated by caspase-induced apoptosis, which causes germ cells to shrink further. In addition, RAS/MAPK signaling may also directly induce apoptosis through unknown mechanisms (dashed arrow). Rachis bridges are symbolized with green circles, cells that decrease in size are shown in pink and round, and apoptotic cells in magenta color.

We have previously reported an interaction between the ECT/RHO and the RAS/MAPK pathways during vulval fate specification in the hermaphrodite larva, where hyperactivation of RHO-1 signaling through the *ect-2(gf)* mutation promotes 1° fate specification upstream of or

in parallel with MPK-1 signaling [23]. Moreover, activation of the EGFR/RAS/MAPK pathway in the 1˚ vulval precursor cells is necessary for their apical constriction, which initiates the invagination of the vulval epithelium [54].

The redistribution of NMY-2 from the rachis bridges to the apical cortex in the germline of *mpk-1b* mutants likely contributes to the expansion of the rachis bridges. Moreover, the increased recoil after laser incision of the apical germ cell cortex in *mpk-1b* mutants suggests that RAS/MAPK signaling is not necessary for actomyosin contractility, but rather for the localization of myosin at the rachis bridges. The reduced levels of NMY-2 at the rachis bridges after the inactivation of RAS/MAPK signaling could be an indirect consequence of changes in gonad architecture that affect the size or integrity of the rachis bridges. Alternatively, RAS/MAPK signaling may directly regulate myosin mobility or recruitment to the rachis bridges, for example, via phosphorylation of the myosin heavy chain NMY-2, which prevents the assembly of myosin into filaments [62]. Such a mechanism has recently been discovered during somatic cell death, where PIG-1 MELK phosphorylates NMY-2 to cause asymmetrical cell division and apoptosis [63,64].

The changes in gonad architecture after hyperactivation or inhibition of the RAS/MAPK pathway (for example, the enlarged rachis in *mpk-1(lf)* mutants) cannot intuitively be explained by local changes in actomyosin contractility alone, suggesting that RAS/MAPK signaling affects gonad morphology through additional mechanisms. For example, inhibition of RAS/MAPK signaling may increase the hydrostatic pressure in the gonad due to the lack of oocyte maturation and ovulation [60] and thereby enlarge the rachis diameter, while elevated RAS/MAPK signaling may result in a lowered hydrostatic pressure and, consequently, a smaller rachis diameter.

Taken together, our data suggest that the RAS/MAPK and ECT/RHO/ROCK pathways act through distinct mechanisms to control germ cell size and death. This model is consistent with previous reports showing that the size of the rachis bridges decreases in the pachytene region where the RAS/MAPK pathway is active and again increases in surviving germ cells that enter the loop region [19]. It should be noted, however, that our data do not exclude the possibility that RAS/MAPK signaling also regulates germ cell death independently of germ cell size through an unknown mechanism (dashed arrow in **Fig 7**).

## A contractility-based model for physiological germ cell death

We propose that actomyosin contractility of the rachis bridges, regulated by the combined actions of the RAS/MAPK and ECT/RHO/ROCK pathways, enhances the intrinsic disparities in cell size in the syncytial gonad region. Germ cell tracking experiments revealed that stochastic, initially small size differences between individual germ cells are rapidly amplified, as slightly smaller cells progressively shrink, while their neighbors increase in size. This phenomenon can be explained by the Law of Laplace, which is illustrated by an experiment with 2 corresponding balloons of unequal sizes, where the initially smaller balloon empties into the larger one. Such a model has been proposed by Chartier and colleagues [32], who found that hydraulic instability in the syncytial germline amplifies stochastic size differences between the germ cells, causing smaller cells to expel their cytoplasm and die. Specifically, the Law of Laplace ($T = P * R/2$, where T = surface tension; P = internal pressure, R = cell radius) predicts that for a given internal pressure, which should be equal in all syncytial germ cells connected to the rachis, the surface tension will be lower, and, therefore, actomyosin contractility can be more effective in smaller cells. Hence, germ cells that are slightly smaller than their neighbors are prone to shrink and expel their cytoplasm until their rachis bridges are closed. According to this model, increasing the activity of the RAS/MAPK or ECT/RHO/ROCK pathway will

enhance actomyosin contractility at the rachis bridges, which will exacerbate this inherent instability and result in even greater cell size discrepancies and an increased rate of germ cell death.

Shrinking germ cells may not only donate their cytoplasm to the survivors but also lose survival factors that prevent the activation of the CED-3 caspase. Preapoptotic germ cells rapidly expel most of their mitochondria [33], which carry antiapoptotic factors such as the BCL-2 homolog CED-9 [65]. However, in mutants blocking mitochondrial expulsion, the germ cells can still undergo apoptosis [33], suggesting that additional antiapoptotic factors localized in the cytoplasm may be lost in small germ cells.

The regulation of physiological germ cell death by cell size probably serves to maintain germline homeostasis and ensure the reallocation of resources from the dying to the surviving germ cells that grow in size while initiating oogenesis [13,15]. The coupling of nutrient signals sensed by the DAF-2 InsR to the RAS/MAPK pathway could be a mechanism that permits the animals to adapt the equilibrium between oocyte differentiation and germ cell death to changing environmental conditions [29].

Cell shrinkage is a hallmark of cells undergoing apoptosis [37]. In most cases, the size reduction has been seen as a consequence rather than a cause of apoptotic cell death. Our findings that actomyosin-induced cell constriction is one factor in selecting smaller germ cells to undergo apoptosis could point to a more widely used mechanism. Even though the *C. elegans* hermaphrodite gonads with their syncytial architecture in the distal region represent a special case, it is conceivable that actomyosin-mediated cell shape changes, for example, during asymmetric cell division, could contribute to the elimination of the usually smaller daughter cells [64,66].

## Materials and methods

### *C. elegans* culture and maintenance

*C. elegans* strains were maintained at 20˚C, unless noted otherwise, on standard nematode growth medium (NGM) agar plates as described [67]. The wild-type N2 strain was *C. elegans*, variety Bristol. We refer to translational protein fusions with a :: symbol between the gene and the tag used, and to transcriptional fusion with a > between the enhancer/promoter used and the gene of interest. Details on the construction of the plasmid vectors can be found in **S2 Data**. The genotypes of the strains, the plasmid vectors, oligonucleotides, and sgRNAs used in this study are listed in **S3 Data**.

### Generation of alleles by CRISPR/CAS9 genome editing

The endogenous *ect-2(zh135)* and *let-502(zh139) gfp* insertions were generated using the modified CRISPR/CAS9 protocol described by [68]. The GFP tag was inserted at the C-terminus of *ect-2* and the N-terminus of *let-502*. The *mpk-1b(zh164)* allele introducing a stop codon at position 4 was generated by the co-CRISPR protocol described in [1].

### Generation of extrachromosomal arrays by microinjection

Microinjection for the generation of the extrachromosomal line (*mpk-1(ga111^{ts}) unc-79 (e1068)* III; *bcIs39[Plim-7::ced-1::gfp]; zhEx676[mpk-1A(+)]* V) was performed as described in [69] using purified PCR DNA at a concentration of 35 ng/μl and the coinjection marker pCFJ90 (P*myo-2*>mCherry) at a concentration of 2.5 ng/μl [70]. pBluescript-KS was added to achieve the final DNA concentration of 150 ng/μl in a total volume of 20 μl. Primary transformants were identified by the mCherry signal from the pharynx.

## RNA interference

RNAi was done by feeding dsRNA-producing *E. coli* [71]. dsRNA-producing bacterial clones targeting genes of interest were obtained from the *C. elegans* genome-wide RNAi library or the *C. elegans* open reading frame (ORFeome) RNAi library (both from Source BioScience). Bacteria were grown in 2 mL of 2xTY medium, containing 200 μg/mL ampicillin and 25 μg/mL tetracycline at 37˚C and either directly seeded on NGM plates containing 3 mM IPTG or diluted into fresh 2xTY medium containing 200 μg/mL ampicillin and 25 μg/mL tetracycline and 1 mM IPTG and grown for 4 hours at 37˚C before seeding [72]. For constitutive and tissue-specific RNAi, larvae were synchronized at the L1 stage by hypochlorite treatment of gravid adults and plated on NGM plates containing dsRNA-expressing bacteria. P0 animals were analyzed after 72 to 74 hours of treatment (1-day-old adults). For transient RNAi, synchronized L1 larvae were grown for 48 hours on OP50 bacteria and transferred as L4 larvae to the RNAi plates for 24 hours.

## Microscopy

Fluorescent and DIC (Nomarski) images were acquired on an Olympus BX61 microscope equipped with an X-light V2 spinning disk confocal system (50 μm pinhole diameter), Prizmatix UHP-T-460-DI/UHP-T-560-DI LEDs as light source, an Andor iXon Ultra 888 EMCCD camera and a 60× or 100× Plan Apo lens (N.A. 1.3 and 1.4, respectively) or on a Leica DMRA microscope equipped with a Lumencor Spectra light engine, 2 Hamamatsu C11400-42U30 camera, a beam splitter, and a 63× Plan Apo lens (N.A. 1.3). Long-term time-lapse recordings were acquired on a Nikon Ti-U microscope equipped with an Omicron LedHUB as a light source, a Photometrics Prime 95B camera, and a PLAN Apo Lambda 60× oil immersion objective. Z-stacks were recorded with a spacing of 0.1 to 0.5 μm depending on the magnification used. For germ cell tracking experiments, 1-day-old adult hermaphrodites were immobilized in custom-made microfluidic devices as described [12], and 41 z-slices were recorded every 7.5 minutes over 12 hours. See the **extended methods** in **S1 File** for a detailed description of the image processing workflow.

## Scoring the apoptotic cell numbers

Cell corpses were counted using strains containing the *bcIs39[Plim-7::ced-1::gfp]* reporter by observing CED-1::GFP expression to identify corpses engulfed by the somatic sheath cells [14,38]. Corpses were counted in 1-day-old adults 72 hours after L1 synchronization at 20˚C or, where indicated, worms were incubated for 68 hours at 20˚C and shifted to 25˚C for 4 hours or 24 hours before quantification.

## Quantification of ERK phosphorylation by western blot analysis

One-hundred 1-day-old adult animals, grown from the L1 stage for 72 hours at 20˚C or incubated for the last 4 or 24 hours at 25˚C where indicated, were washed 3 times in ice-cold M9, collected by centrifugation, and lysed in 50 μl 1xSDS samples buffer for 5 minutes at 95˚C. To remove genomic DNA, 1 μl of RNase-Free DNase (QIAGEN) was added and the samples were incubated at room temperature for 10 minutes, and again for 5 minutes at 95˚C immediately before loading. Twenty μl extract each were loaded on two 4% to 12% gradient polyacrylamide gels (Invitrogen) run in parallel. Western blots were incubated with anti-diphospho or total MAP Kinase antibodies as well as anti-alpha tubulin antibodies as loading controls. Bound primary antibodies were detected with HRP-conjugated secondary antibodies followed by a chemiluminescence assay (SuperSignal West Dura Extended Duration Substrate, Thermo

Scientific). To inactivate HRP conjugates and proceed with the incubation of the primary anti-alpha tubulin antibody, the western blots were incubated in sodium azide for 1 hour. Bound primary antibodies were detected with HRP-conjugated secondary antibody and chemiluminescence assay (SuperSignal West Pico PLUS Chemiluminescent Substrate, Thermo Scientific). Total and diphospho MPK-1B and alpha-tubulin levels were quantified by measuring the corresponding band intensities using built-in measurement tools of Fiji [73]. Diphosphorylated MPK-1B and total MPK-1B intensities were normalized to the alpha-tubulin signals as loading controls before the diphospho to total MPK-1B ratios were calculated. The ratios shown in the graph **Fig 6C'** are the average values of 3 biological replicates (2 for *let-502 (lf)*), normalized in each experiment to the rations measured for the wild-type controls. Antibodies used are as follows: anti-diphosphoERK, activated (Sigma-Aldrich, M8159); anti-MAP Kinase (Sigma-Aldrich, M5670), anti-alpha tubulin (Abcam, ab18251), anti-alpha tubulin (Sigma-Aldrich, T6074), HRP anti-rabbit (Jackson Immuno Research, 111-035-144), and HRP anti-mouse (Jackson Immuno Research, 115-035-146).

### Laser incision and recoil measurements

Laser incisions were generated using an Olympus IXplore SpinSR10 microscope equipped with a 355-nm pulsed laser for photomanipulation (UGA-42 Caliburn, Rapp Optoelectronics), a z-drift compensator (IX3-ZDC2, Olympus) and a Hamamatsu ORCA-Fusion sCMOS camera, using a 60× Plan UPlan Apo Silicone oil immersion lens (N.A. 1.3). Incisions were made with 1,000 ms laser pulses at 8% to 10% laser power. Animals mounted on agarose pads were observed for 2 frames before and 60 frames after the incision at a rate of 1.33 frames per second. For quantification, 3 to 4 bright NMY-2::GFP foci around the incision point were manually tracked over 40 frames using Fiji software [73] to measure their relative displacement from the point of incision, from which the average radial displacement (recoil) for each animal and time point was calculated (**S1 Data**). Curve-fitting was done individually for each animal recorded using the equation

$$recoil(t) = initial\ recoil\ velocity/k*(1-e^{-k*t})$$

to estimate the initial recoil velocity (= $\partial recoil(0)/\partial t$) and the total recoil (recoil($\infty$) = initial recoil velocity/k, where k = elasticity of the tissue/viscosity coefficient of the drag in the cytoplasm) using Graph Pad Prism software as described [53]. Since we quantified the radial displacement rather than the change in diameter of the opening, the values obtained are approximately half of those reported by [18].

### Statistical analysis

GraphPad Prism 9.0 was used to perform statistical tests, and results are shown in **S1 Data**. We used two-tailed unpaired *t* tests when comparing 2 samples or ANOVA with Tukey's correction for multiple comparisons when comparing more than 2 samples to calculate the *p*-values. Samples sizes (numbers of animals or cells analyzed) are indicated in the figure legends for each experiment. All the experiments were repeated with at least 3 independent biological replicates, except for **Figs 3I** (left), **4A, 4B,** and **4C** with 2 biological replicates).

### Supporting information

**S1 Fig. Related to Fig 1.**
(PDF)

**S2 Fig. Related to Fig 2.**
(PDF)

**S3 Fig. Related to Fig 3.**
(PDF)

**S4 Fig. Related to Fig 4.**
(PDF)

**S5 Fig. Related to Fig 5.**
(PDF)

**S6 Fig. Related to Fig 6.**
(PDF)

**S1 Movie. Germ cell tracking of the wild-type animal shown in Fig 1B.**
(AVI)

**S2 Movie. Germ cell tracking of the *ced-3(lf)* mutant animal shown in Fig 2A.**
(AVI)

**S3 Movie. Recoil after laser incision in the wild-type animal shown in Fig 6E.**
(AVI)

**S4 Movie. Recoil after laser incision in the *mpk-1b(lf)* animal shown in Fig 6E'.**
(AVI)

**S1 Raw images. Uncropped western blots used for quantification in Fig 6B.**
(PDF)

**S1 Data. Statistical tests and underlying data used to generate the graphs.**
(XLSX)

**S2 Data. Details on the construction of the plasmid vectors.**
(XLSX)

**S3 Data. Genotypes of the strains, the plasmid vectors, oligonucleotides, and antibodies used.**
(XLSX)

**S1 File. Extended methods and scripts used.**
(PDF)

## Acknowledgments

We thank all present and past members of the Hajnal group for critical discussion and comments, Michael Daube for lab assistance, and Michael Walser for help with the illustrations. We thank Anne-Lise Routier-Kierzkowska and Richard Smith for their help with using MorphographX software. We are also grateful to Barbara Conradt for sharing the MD4476 strain and for helpful comments, to Patrick Narbonne for the UTR137 strain, to Andrew Fire for vectors, to J. Ahringer for RNAi clones, and to WormBase. We thank the team of the Center for Microscopy and Image Analysis, University of Zurich, for their assistance with performing the laser cutting experiments. Some strains were provided by the CGC, which is funded by the NIH Office of Research Infrastructure Programs (P40 OD010440).

## Author Contributions

**Conceptualization:** Tea Kohlbrenner, Alex Hajnal.

**Formal analysis:** Tea Kohlbrenner, Simon Berger, Tinri Aegerter-Wilmsen.

**Funding acquisition:** Tea Kohlbrenner, Andrew deMello, Alex Hajnal.

**Investigation:** Tea Kohlbrenner, Simon Berger, Ana Cristina Laranjeira, Laura Filomena Comi, Alex Hajnal.

**Writing – original draft:** Tea Kohlbrenner, Alex Hajnal.

**Writing – review & editing:** Tea Kohlbrenner, Simon Berger, Ana Cristina Laranjeira, Tinri Aegerter-Wilmsen, Andrew deMello, Alex Hajnal.

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
