## [Editor Report · Decision Letter 0]

7 Sep 2023

Dear Dr Hajnal, 

Thank you for submitting your manuscript entitled "Actomyosin-mediated apical constriction promotes physiological germ cell death in C. elegans" for consideration as a Research Article by PLOS Biology.

Your manuscript has now been evaluated by the PLOS Biology editorial staff as well as by an academic editor with relevant expertise and I am writing to let you know that we would like to send your submission out for external peer review.

Once your full submission is complete, your paper will undergo a series of checks in preparation for peer review. After your manuscript has passed the checks it will be sent out for review. To provide the metadata for your submission, please Login to Editorial Manager (https://www.editorialmanager.com/pbiology) within two working days, i.e. by Sep 11 2023 11:59PM.

Kind regards,

Ines

--

Ines Alvarez-Garcia, PhD

Senior Editor

PLOS Biology

---

## [Decision Letter · Decision Letter 1]

15 Nov 2023

Dear Dr Hajnal,

Thank you for your patience while your manuscript entitled "Actomyosin-mediated apical constriction promotes physiological germ cell death in C. elegans" was peer-reviewed at PLOS Biology. It has now been evaluated by the PLOS Biology editors, an Academic Editor with relevant expertise, and by two independent reviewers. 

As you will see, the reviewers find the topic and conclusions interesting for the field, however they have also raised several concerns that would need to be addressed before we consider the manuscript for publication. While Reviewer 1 mostly asks for several clarifications, Reviewer 2 is not convinced that the experiments performed support the model proposed. This reviewer suggests further experiments that should be done to confirm the conclusions and also an alternative interpretation of the results that should be tested. In addition, given that binucleate germ cells are eliminated by apoptosis, you should also quantify the number of these cells after the cytoskeletal perturbations, and also be cautious when drawing interpretations using gain-of-function mutations.

In light of the reviews, we would like to invite you to revise the work to thoroughly address the reviewers' reports.

Given the extent of revision needed, we cannot make a decision about publication until we have seen the revised manuscript and your response to the reviewers' comments. Your revised manuscript is likely to be sent for further evaluation by all or a subset of the reviewers.

**IMPORTANT - SUBMITTING YOUR REVISION**

3. Resubmission Checklist

a) *PLOS Data Policy*

b) *Published Peer Review*

Sincerely,

Ines

--

Ines Alvarez-Garcia, PhD

Senior Editor

PLOS Biology

Reviewers' comments

Rev. 1:

Summary: Programmed cell deaths play a vital part in the development and homeostasis of all animals. This role is particularly important in the female germ line. Although the genetic control of cell death was first described in the nematode C. elegans decades ago, the regulation of germ cell deaths in this animal is complex and remains confusing. Here the authors combine a wide array of experimental techniques to support the model that smaller germ cells are preferentially eliminated by programmed cell deaths late in the pachytene phase of meiosis I, and that their small size appears to determine cell death, rather than being an early cause of it. They dissect the regulation of germ cell size by the ECT/RHO/ROCK and RAS/MAPK pathways, and show that these genes control the constriction of the actomyosin network that determines the size of the pores connecting each germ cell to the central core of the syncytium. These results are an important advance for understanding cell death in nematodes and in germ cells, and could have much broader implications.

Recommendation: This is a beautiful, detailed study of broad interest to readers of PLoS Biology. It is based on extensize and sophisticated experiments, and the arguments unfold in a very logical and organized manner. I strongly recommend publication.

Comments:

(1) The mpk-1b phenotype is so large in figures 4B and 6A, that I wonder if the somatic effects of MPK-1a not only involve corpse clearing, but also the decision to die. It would be very helpful for the reader if the authors would extend their description of the mpk-1b mutant to a survey of all germline and somatic gonad phenotypes, so the reader could consider whether other big interactions might also being altered.

(2) In lines 611-612, the authors write "In the C. elegans hermaphrodite germline, RAS/MAPK

signaling plays an essential role at two stages…" However, Lee et al 2007 identified four roles for mpk-1 in the germ line.

(3) When the authors first mention 2.5D projections, they should describe them briefly.

Corrections

Line 40 Change "before they can enter" to "before they can complete"

Line 51 Change "1'000" to "1000"

Line 151 Change "are also labeled in black." to "are also labeled in gray."

Line 312 Change "let-60 ras(ga89ts)" to "let-60 ras(ga89ts)"

Line 446 Change "mpk-1(ts lf)" to "mpk-1(lf ts)"

Line 466 Change "mpk-1(gf ts)" to "mpk-1(lf ts)"

Line 615 Change "the membrane integrity," to "membrane integrity,"

Rev. 2: David Greenstein – note that this reviewer has signed his review.

Summary

Programmed cell death or apoptosis is an important cell fate in developmental biology, needed to sculpt tissues and organs. A key question is how cells adopt the apoptotic fate. Studies in C. elegans have pioneered the field. In the C. elegans soma, expression of the proapoptotic BH3-only protein EGL-1 is a major determinant of the apoptotic cell fate choice. By contrast, in the C. elegans germline in which ~60-80% of oogenic germ cells undergo apoptosis, EGL-1 is dispensable for the apoptotic cell fate choice. How female germ cells are allocated to this cell fate is an unanswered question in the field. This study from the Hajnal lab presents a new model in which the RAS/MAPK pathway functions in parallel to an ECT/RHO/ROC actin regulatory pathway upstream of CED-3 Caspase activation to allocate female germ cells to the apoptotic pathway (see their Figure 7).

Critique

I am not convinced by the data in support of the authors' model. Discrepancies with prior publications have not been adequately addressed in the manuscript. Physiological cell death in the C. elegans germline proceeds in three phases: induction, execution, and clearance. The shrinkage of apoptotic germ cells has been well documented in prior studies. This actin-dependent shrinkage of germ cells is generally thought of as being a key part of the execution phase. Importantly, the earliest reported phene of apoptotic germ cells is that they export mitochondria to the germline core cytoplasm (rachis) in a microtubule-dependent manner (Raiders SA, Eastwood MD, Bacher M, Priess JR. Binucleate germ cells in Caenorhabditis elegans are removed by physiological apoptosis. PLoS Genet. 2018 Jul 19;14(7):e1007417. doi: 10.1371/journal.pgen.1007417. PMID: 30024879; PMCID: PMC6053125.). This early mitochondrial export is then followed by the shrinkage of germ cells and their engulfment by the somatic sheath cells. In their studies, the authors have not examined the earliest stages of the apoptotic process—mitochondrial export. An alternative possibility is that the experimental perturbations are augmenting the execution phase.

The experiments presented in the manuscript are not sufficient to position the ECT/RHO/ROC actin regulatory pathway upstream of CED-3. It is also possible that the experimental treatments cause syncytial germ cells that are near the threshold of CED-3 caspase activation to commit to the apoptotic cell fate. In this manner, a downstream step in the execution step might feedback on the decision. There is precedent for such a mechanism in the C. elegans soma. For example, Reddien et al. (2001) showed that engulfment, which is a step downstream of caspase activation, could promote apoptosis when ced-3 activity was limiting. (Reddien PW, Cameron S, Horvitz HR. Phagocytosis promotes programmed cell death in C. elegans. Nature. 2001 Jul 12;412(6843):198-202. doi: 10.1038/35084096. PMID: 11449278). As the authors acknowledge, their experimental perturbations of both the actin regulatory pathways and the RAS/MAPK pathway are highly pleiotropic to the germline.

The authors themselves mention that the role of the RAS/MAPK pathway in physiological germ cell death is "controversial" (line 307). The manuscript inadequately presents this controversy and fails to resolve the issue. It is known that strong loss-of-function mpk-1 alleles abolish germline apoptosis; however, germ cells in these mutants arrest in the mid-pachytene stage. Since apoptosis occurs mainly among more proximal pachytene cells, it might be that these germ cells have not progressed to the stage of pachytene in which they undergo apoptosis. In fact, when mpk-1(ga111ts) mutants are shifted to the non-permissive temperature, an increased number of apoptotic germ cell corpses is observed (Arur S, Ohmachi M, Nayak S, Hayes M, Miranda A, Hay A, Golden A, Schedl T. Multiple ERK substrates execute single biological processes in Caenorhabditis elegans germ-line development. Proc Natl Acad Sci U S A. 2009 Mar 24;106(12):4776-81. doi: 10.1073/pnas.0812285106. Epub 2009 Mar 5. PMID: 19264959; PMCID: PMC2660749. See their Figure S1). The authors should cite this paper and more completely investigate whether mpk-1 promotes or inhibits (or both) the apoptotic fate using mpk-1(ga111ts) mutants and other approaches. As it is, the manuscript over-relies on using a weak gain-of-function let-60/ras mutation. The problem is that one might imagine the possibility that stronger (or weaker) gain-of-function mutations might produce different outcomes.

Major Points

1. More convincing evidence is needed to support the idea that apical constriction functions in the induction phase rather than the execution phase.

2. The manuscript should more completely address and resolve the issue of the role of mpk-1 in germline apoptosis.

3. In their Abstract, the authors make the claim that "Through this mechanism, animals can adapt the rate of germ cell death and differentiation to changing environmental conditions." I could not find data in the manuscript supporting this claim.

4. Lines 83-84. This statement furthers a misperception in the field. As discussed above, it may be that mpk-1 mutant female germ cells do not progress to the stage in which they undergo programmed cell death. In fact, there is evidence that MPK-1 might inhibit apoptosis.

5. The authors might consider analyzing a partial loss-of-function ced-3 allele, ced-3(n2427), to determine whether they might be able to enhance or suppress its effects by modulating the pathways they describe.

6. The authors analyze ect-2(gf); cep-1(lf) mutants and ect-2(gf); pch-2(lf) mutants. Given that perturbations of the actin cytoskeleton are pleiotropic, both pathways might be engaged. Thus, the analysis should include ect-2(gf) pch-2(lf); cep-1(lf) triple mutants.

7. Lines 328-329 and subsequent text. The manuscript inadequately addresses the possibility that mpk-1a functions in sheath cells to regulate engulfment. Although the generation of a germline-specific mpk-1 allele is useful, it doesn't resolve the possibility that the absence of apoptotic germ cells when mpk-1 activity is depleted from the germline is a secondary consequence of the meiotic progression defect. The authors might also examine ect-2(gf); mpk-1(ga111ts) mutants in addition to mpk-1(ga111ts) single mutants.

8. Lines 574-576. The authors state, "Here, we show that pachytene stage germ cells smaller than their neighbors are selectively eliminated through apoptosis, probably to make space for and donate their resources to the surviving germ cells entering oogenesis." If, as prior work indicates, germ cell shrinkage is a key element of the execution phase (see Raiders et al., 2018), the reason the smaller germ cells are eliminated is that they are smaller because they have already adopted the apoptotic fate and begun to shrink. If so, the expectation would be that the smaller germ cells had already undertaken the first known step of the apoptotic pathway in which mitochondria exit from the germ cells into the rachis prior to shrinking.

Minor Points

1. Line 40-41. The apoptotic germ cells have already adopted a female fate. So, it would be preferable to state something like, "before they differentiate as oocytes."

2. Lines 44-46. Physiological apoptosis also removes binucleate germ cells (Raiders SA, Eastwood MD, Bacher M, Priess JR. Binucleate germ cells in Caenorhabditis elegans are removed by physiological apoptosis. PLoS Genet. 2018 Jul 19;14(7):e1007417. doi: 10.1371/journal.pgen.1007417. PMID: 30024879; PMCID: PMC6053125).

3. Lines 46-49. This statement should be modified because egl-1 is dispensable for physiological germ cell apoptosis.

4. Line 51. 1,000.

5. Lines 79-80. Sustained MPK-1 activation in pachytene requires the presence of sperm (Lee MH, Ohmachi M, Arur S, Nayak S, Francis R, Church D, Lambie E, Schedl T. Multiple functions and dynamic activation of MPK-1 extracellular signal-regulated kinase signaling in Caenorhabditis elegans germline development. Genetics. 2007 Dec;177(4):2039-62. doi: 10.1534/genetics.107.081356. PMID: 18073423; PMCID: PMC2219468.)

---

## [Decision Letter · Decision Letter 2]

24 Jun 2024

Dear Dr Hajnal,

Thank you for your patience while we considered your revised manuscript entitled "Actomyosin-mediated apical constriction promotes physiological germ cell death in C. elegans" for publication as a Research Article at PLOS Biology. This revised version of your manuscript has been evaluated by the PLOS Biology editors, the Academic Editor and one of the original reviewers.

Based on the review, we are likely to accept this manuscript for publication, provided you satisfactorily address the data and other policy-related requests stated below.

We expect to receive your revised manuscript within two weeks. 

*Published Peer Review History*

*Press*

Sincerely,

Ines

--

Ines Alvarez-Garcia, PhD

Senior Editor

PLOS Biology

Fig. 1F, G, K, O; Fig. 2A, B; Fig. 3D, G, H, I; Fig. 4A-D, I, J; Fig. 5C, F-I; Fig. 6A, B, F-H; Fig. S1A, B, D, E; Fig. S2A, B; Fig. S3G, H, J’; Fig. 4A-F; Fig. 5E and Fig. S6B-G

CODE POLICY

We require the original, uncropped and minimally adjusted images supporting all blot and gel results reported in an article's figures or Supporting Information files. We will require these files (in Fig. S6A-A’’) before a manuscript can be accepted so please prepare and upload them now. Please carefully read our guidelines for how to prepare and upload this data: https://journals.plos.org/plosbiology/s/figures#loc-blot-and-gel-reporting-requirements

Reviewers' comments

Rev. 2: David Greenstein

The authors have provided a revision that satisfactorily addresses the majority of the points raised in the prior critiques.

---

## [Editor Report · Decision Letter 3]

30 Jul 2024

Dear Dr Hajnal,

Thank you for the submission of your revised Research Article entitled "Actomyosin-mediated apical constriction promotes physiological germ cell death in C. elegans" for publication in PLOS Biology. On behalf of my colleagues and the Academic Editor, Yukiko Yamashita, I am delighted to let you know that we can in principle accept your manuscript for publication, provided you address any remaining formatting and reporting issues. These will be detailed in an email you should receive within 2-3 business days from our colleagues in the journal operations team; no action is required from you until then. Please note that we will not be able to formally accept your manuscript and schedule it for publication until you have completed any requested changes.

PRESS

Sincerely, 

Ines

--

Ines Alvarez-Garcia, PhD

Senior Editor

PLOS Biology
